# Understanding Diffusion-based Representation Learning via Low-Dimensional Modeling

## Abstract

This work addresses the critical question of why and when diffusion models, despite their generative design, are capable of learning high-quality representations in a self-supervised manner. We hypothesize that diffusion models excel in representation learning due to their ability to learn the low-dimensional distributions of image datasets via optimizing a noise-controlled denoising objective. Our empirical results support this hypothesis, indicating that variations in the representation learning performance of diffusion models across noise levels are closely linked to the quality of the corresponding posterior estimation. Grounded on this observation, we offer theoretical insights into the unimodal representation dynamics of diffusion models as noise scales vary, demonstrating how they effectively learn meaningful representations through the denoising process. We also highlight the impact of the inherent parameter-sharing mechanism in diffusion models, which accounts for their advantages over traditional denoising auto-encoders in representation learning.

## 1 Introduction

Diffusion models, a new family of likelihood-based generative models, have demonstrated superior performance among many generative tasks, including image generation (Alkhouri et al., 2024; Ho et al., 2020; Rombach et al., 2022; Zhang et al., 2024), video generation (Bar-Tal et al., 2024; Ho et al., 2022), speech and audio synthesis (Kong et al., 2020; 2021), semantic editing (Roich et al., 2022; Ruiz et al., 2023; Chen et al., 2024a) and solving inverse problem (Chung et al., 2022; Song et al., 2024; Li et al., 2024; Alkhouri et al., 2023). At its core, diffusion models are learning a data distribution from training samples by imitating the non-equilibrium thermodynamic diffusion process (Sohl-Dickstein et al., 2015; Ho et al., 2020; Song et al., 2021). In the forward process, training samples are gradually combined with increasing Gaussian noise until the data structure is completely destroyed while in the backward process, a model is trained to restore the structure from the noised data (Hyvärinen & Dayan, 2005; Song et al., 2021).

In addition to their impressive generative capabilities, recent studies (Baranchuk et al., 2021; Xiang et al., 2023; Mukhopadhyay et al., 2023; Chen et al., 2024b; Tang et al., 2023) have highlighted the exceptional representation power of diffusion models, suggesting that they could serve as a unified foundation model for both generative and discriminative vision tasks. Specifically, recent evaluations across various applications, including classification (Xiang et al., 2023; Mukhopadhyay et al., 2023), semantic segmentation (Baranchuk et al., 2021), and image alignment (Tang et al., 2023), show that diffusion models are capable of learning high-quality representations, often matching or even surpassing the performance of previous state-of-the-art methods. However, it remains unclear whether the representation capabilities of diffusion models stem from the diffusion process or the denoising mechanism (Fuest et al., 2024). More fundamentally, given their generative design, *when and why diffusion models can learn high-quality representations in a self-supervised manner?*

This work aims to address this question through a comprehensive investigation, both empirically and theoretically, grounded in the formulation of denoising auto-encoders (DAEs) for learning diffusion models (Vincent et al., 2008; 2010; Vincent, 2011). We hypothesize that diffusion models can learn high-quality representations without supervision due to their superior ability to approximate the low-dimensional distributions of image datasets, as supported by recent findings (Wang et al., 2024). Although image dataset can be very high-dimensional, recent results (Pope et al., 2021;

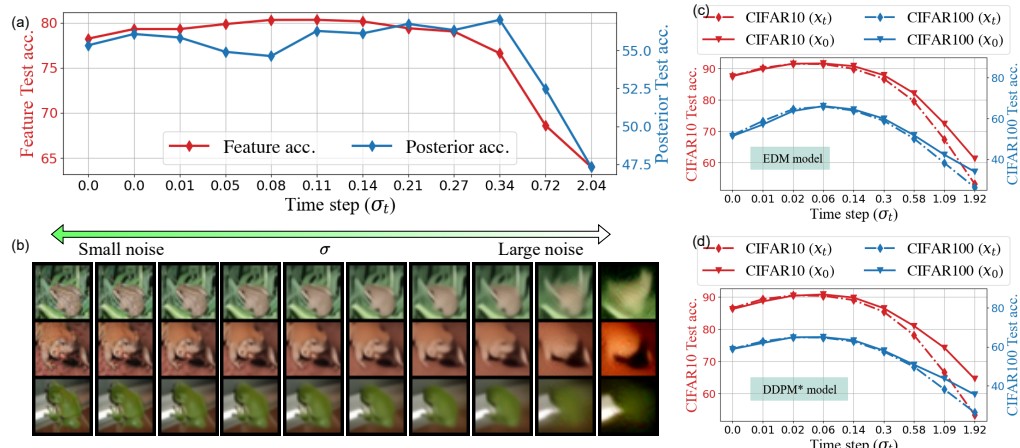

Figure 1: **Representation learning ability of a diffusion model at different time steps reflects the granularity in posterior estimation.** (a) Intermediate feature posterior probing accuracy of the diffusion model exhibit a similar unimodal trend as noise level increases. (b) Posterior estimation for clean image inputs shows a transition from fine to coarse granularity with increasing noise levels. (c)-(d) Using clean image input $x_0$ for feature extraction achieves comparable or superior representation learning performance compared to using noisy input $x_t$.

Stanczuk et al., 2022; Wang et al., 2024) demonstrate that the intrinsic dimension of these datasets are much lower than the ambient dimension, and it has shown that the number of samples to learn the underlying distribution using diffusion models scales with the intrinsic low-dimensionality. Therefore, by being trained to capture the underlying structure of data through a controlled process of noise injection and denoising, diffusion models effectively learn meaningful and compact features.

On the empirical side, we support our claim by reconciling several intriguing phenomena related to the quality of learned representations in diffusion models. Recent studies Zhang et al. (2023) reveal that diffusion models operate in two regimes: memorization and generalization, depending on training data size. In the memorization regime with limited samples, the model captures only the empirical distribution of training data without the ability to generate new samples. In contrast, in the generalization regime, diffusion models are able to learn the underlying distribution. Our experiments in Figure 2 confirm that high-quality representations are *only* learned in the generalization regime with sufficient samples due to its ability of learning the underlying distribution. More importantly, in the generalization regime, we show that the quality of hidden representations in diffusion models/DAEs follows a uni-modal curve (see Figure 1 and Figure 7): high-quality representations are learned at an intermediate step close to the clean image, whereas the representation quality degrades as it approaches either pure noise or the clean image.

Building on these empirical observations, we provide theoretical insights using a noisy mixture of low-rank Gaussian distributions. Our assumption captures the inherent low-dimensionality of the image data distribution (Pope et al., 2021; Gong et al., 2019; Stanczuk et al., 2022), where the data lies on a union of low-dimensional subspaces. We analyze the unimodal trend in representation performance by relating it to the Class-specific Signal-to-Noise Ratio (CSNR). Specifically, we consider the optimal posterior estimation function under our data assumption and show that the CSNR is determined by the interplay between data "denoising" and class confidence rate as the noise scale increases. Additionally, our study reveals an implicit weight-sharing mechanism inherent in diffusion models, which helps explain their strengths compared to traditional one-step DAEs, particularly in the small noise regions.

**Contribution of this work.** In summary, our findings can be highlighted as follows:

- **Linking posterior estimation ability of diffusion models to representation learning.** Our empirical results reveal that, much like the dynamics of diffusion representation learning, posterior estimation quality across noise levels follows a similar unimodal curve. This indicates that changes in representation quality are a direct reflection of changes in posterior estimation quality, prompting us to explore representation learning through the more fundamental lens of posterior recovery.

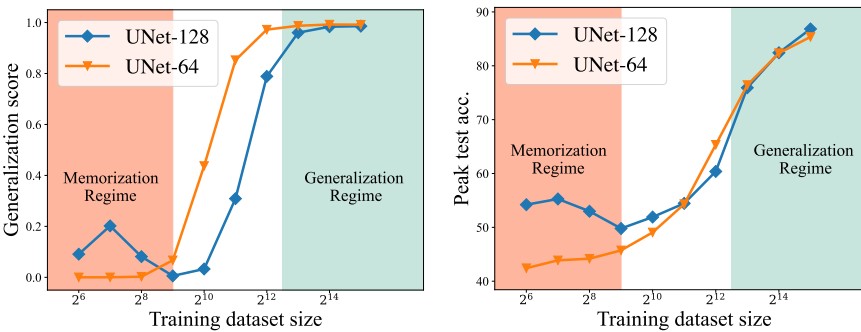

(a) Phase transition in generalization score    (b) Phase transition in representation learning

Figure 2: **Better representations are learned in the generalization regime.** We train EDM-based (Karras et al., 2022) diffusion models on the CIFAR-10 dataset using different training dataset sizes, ranging from $2^6$ to $2^{15}$. (a) The change in the generalization score (Zhang et al., 2023) as the dataset size increases, where regions with a generalization score close to 0 are labeled as the memorization regime, and those close to 1 are labeled as the generalization regime. (b) The peak representation learning accuracy achieved as a function of dataset size.

- **Theoretical analysis of the unimodal curve in the denoising process.** Building on the connection between posterior estimation and representation learning, we present the first theoretical framework for analyzing the unimodal evolution of representation quality. Using a mixture of low-rank Gaussian data model, we demonstrate that the unimodal curve arises from the interplay between denoising strength and class confidence as the noise level varies.
- **Weight sharing in the diffusion process.** Furthermore, we reveal that the diffusion process, by minimizing losses across all noise levels simultaneously, fosters an implicit parameter sharing mechanism within a diffusion model. This mechanism plays a crucial role for diffusion models to achieve superior and more consistent representation learning performances compared with traditional DAEs.

## 2 REPRESENTATION LEARNING VIA DIFFUSION MODELS

In this section, we first review the fundamentals of diffusion models and outline the feature extraction method used in this work. Following this, we illustrate the connection between diffusion posterior estimation and representation learning, which serves as the foundation for the subsequent analysis in Section 3.

### 2.1 PRELIMINARIES ON DENOISING DIFFUSION MODELS

Diffusion models are a class of probabilistic generative models that aim to reverse a progressive noising process by mapping an underlying data distribution, $p_{\text{data}}$, to a Gaussian distribution.

**The forward process.** Starting from clean data $\boldsymbol{x}_0$, noise is gradually introduced according to a noise schedule determined by the time step $t$ until the data becomes indistinguishable from pure Gaussian noise. Specifically, at any time step $t$, the noised data can be expressed as: $\boldsymbol{x}_t = s_t \boldsymbol{x}_0 + s_t \sigma_t \boldsymbol{\epsilon}$ where $\boldsymbol{\epsilon} \sim \mathcal{N}(\boldsymbol{0}, \boldsymbol{I})$ represents noise sampled from a Gaussian distribution, $s_t$ and $s_t \sigma_t$ represent the scaling of the signal and noise, respectively.

**The reverse process.** Noise is gradually removed from $\boldsymbol{x}_1$ following the reverse-time SDE:

$$\mathrm{d}\boldsymbol{x}_t = \left(f(t)\boldsymbol{x}_t - g^2(t)\nabla \log p_t(\boldsymbol{x}_t)\right)\mathrm{d}t + g(t)\mathrm{d}\bar{\boldsymbol{w}}_t, \tag{1}$$

where $\{\bar{\boldsymbol{w}}_t\}_{t \in [0,1]}$ is the standard Wiener process running backward in time from $t = 1$ to $t = 0$ and the functions $f(t), g(t) : \mathbb{R} \to \mathbb{R}$ respectively denote the drift and diffusion coefficients. Notably, if both $\boldsymbol{x}_1$ and $\nabla \log p_t$ are known, the reverse process mirrors the forward process at each time step $t \geq 0$ (Anderson, 1982).

**Score approximation and denoising auto-encoders (DAEs).** However, the score function $\nabla \log p_t$ is typically unknown, as it depends on the underlying data distribution $p_{\text{data}}$. To address this, a neural network $s_{\boldsymbol{\theta}}$ is trained to estimate the score at various time steps (Ho et al., 2020; Song et al., 2021). Given the relationship between the score function and the posterior mean $\mathbb{E}[\hat{\boldsymbol{x}}_0 | \boldsymbol{x}_t]$

(Vincent, 2011; Wang et al., 2024):

$$s_t \, \mathbb{E}\left[\hat{\boldsymbol{x}}_0 | \boldsymbol{x}_t\right] = \boldsymbol{x}_t + s_t^2 \sigma_t^2 \nabla \log p_t(\boldsymbol{x}_t) \approx \boldsymbol{x}_t + s_t^2 \sigma_t^2 s_{\boldsymbol{\theta}}(\boldsymbol{x}_t), \qquad (2)$$

prior works (Chen et al., 2024b; Xiang et al., 2023; Kadkhodaie et al., 2023) have also proposed an alternative DAE-based training objective that directly estimates the posterior mean $\mathbb{E}[\boldsymbol{x}_0 | \boldsymbol{x}_t]$:

$$\min_{\boldsymbol{\theta}} \; \ell(\boldsymbol{\theta}) := \frac{1}{2N} \sum_{i=1}^{N} \int_0^1 \lambda_t \, \mathbb{E}_{\boldsymbol{\epsilon} \sim \mathcal{N}(\mathbf{0}, \boldsymbol{I}_n)} \left[ \left\| \boldsymbol{x}_{\boldsymbol{\theta}}(s_t \boldsymbol{x}_0^{(i)} + s_t \sigma_t \boldsymbol{\epsilon}, t) - \boldsymbol{x}_0^{(i)} \right\|^2 \right] \mathrm{d}t, \qquad (3)$$

where $\boldsymbol{x}_{\boldsymbol{\theta}}(\boldsymbol{x}_0, t)$ denotes the posterior estimating network, $N$ represents the size of the training dataset, and $\lambda_t$ denotes the weighting for each noise level. To simplify the analysis, we assume throughout the paper that $s_t = 1$ and $\lambda_t$ remain constant across all noise levels, with the noise level denoted as $\sigma_t$.

We note that if we remove the integration in (3) and fix $t$, the loss simplifies to the traditional single-level DAE loss (Vincent et al., 2008), where the DAE is trained at a single noise level. Previous work (Chen et al., 2024b) has decomposed the training objective of diffusion models into the denoising process (through the denoising loss) and the diffusion process (integrating the loss across all noise levels in (3)). To comprehensively investigate the distinct roles of these two processes in representation learning, we consider both diffusion models and individual DAEs in our experiments where the individual DAEs serve as a control group, allowing us to isolate and analyze the effects of the denoising process alone.

## 2.2 EXTRACTING REPRESENTATIONS FROM DIFFUSION MODEL

In this work, we always refer representation quality to the quantitative metrics used in downstream tasks—such as accuracy in classification and adopt the following feature extraction setups to leverage diffusion models for representation learning:

**Use clean images as network inputs.** First, we use the clean image $\boldsymbol{x}_0$ as input to the network in contrast to conventional approaches that use the noisy image $\boldsymbol{x}_t$ (Xiang et al., 2023; Baranchuk et al., 2021; Tang et al., 2023). This setup aligns with the goal of representation learning: when training neural networks for classical representation tasks, whether in a supervised or self-supervised manner, it is standard practice to apply some kind of data augmentations or corruptions—such as cropping, color jittering, or masking. These augmentations improve the robustness of the trained model and enhance performance. However, during inference, clean, unaugmented images are typically used as inputs. Similarly, in diffusion models, since our focus is on their role in representation learning, additive Gaussian noise serves as a form of data augmentation, necessary only during training. During inference, using the clean image $\boldsymbol{x}_0$ as input is sufficient. As demonstrated in Figure 1(c)-(d), this approach preserves the overall unimodal representation dynamic while achieving better performance at higher noise levels. As such, throughout the remainder of this paper, we use the clean data $\boldsymbol{x}_0$ as input to the diffusion model, i.e., we always consider $\boldsymbol{x}_{\boldsymbol{\theta}}(\boldsymbol{x}_0, t)$ where $t$ serves solely as an indicator of the noise level for diffusion model to adopt during feature extraction.

**Layer selection for representations.** Second, we extract features only from the bottleneck layer of the U-Net architecture (Ronneberger et al., 2015),[1] following the protocols used in (Kwon et al., 2022; Park et al., 2023).[2] Unlike prior methods (Xiang et al., 2023; Baranchuk et al., 2021), we do not conduct a grid search for the optimal layer, as our focus is on understanding the process rather than achieving state-of-the-art results.

## 2.3 RELATIONSHIP BETWEEN LEARNED REPRESENTATIONS & POSTERIOR ESTIMATION

**Relationship among posterior estimation, distribution recovery, and representation learning.** Since directly studying representation ability is challenging, in Section 3 we approach the problem through its strong correlation with posterior mean estimation, $\mathbb{E}[\boldsymbol{x}_0 | \boldsymbol{x}_t]$. As we will argue, diffusion representation quality is closely linked with the semantic information encoded in the posterior estimation. Additionally, empirical validations can be found in Figure 1.

- *Posterior estimation and distribution recovery.* Diffusion models are trained to learn the underlying data distribution by reconstructing the posterior mean $\mathbb{E}[\boldsymbol{x}_0 | \boldsymbol{x}_t]$ for a given input $\boldsymbol{x}_t$ at the

---

[1]In other words, the layer with the smallest feature resolution.

[2]After feature extraction, we apply a global average pooling to the features. For instance, given a feature map of dimension $256 \times 4 \times 4$, we pool the last two dimensions, resulting in a 256-dimensional vector.

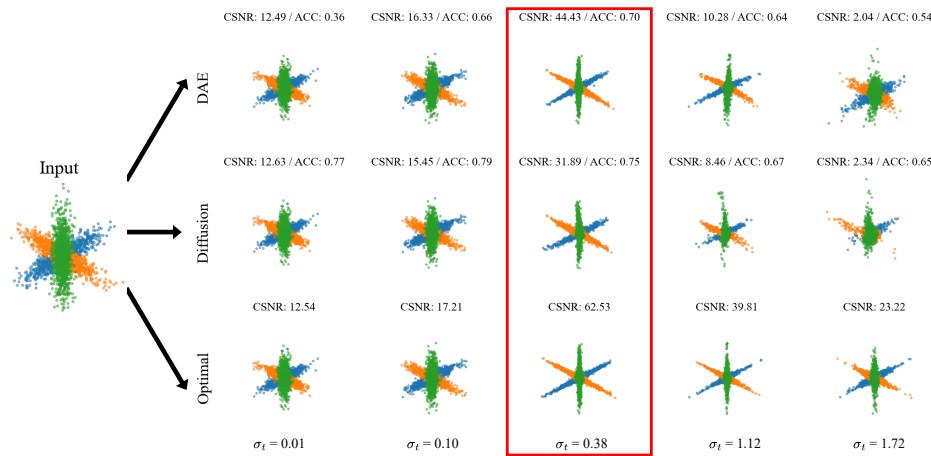

Figure 3: **Visualization of posterior estimation for a clean input.** The **same** `MoLRG` data is fed into the models; each row represents a different denoising model, and each column corresponds to a different time step with noise scale ($\sigma_t$). The red box indicates the best posterior estimation and feature probing accuracy.

specified noise level. Therefore, the quality of posterior estimation $\mathbb{E}[\boldsymbol{x}_0|\boldsymbol{x}_t]$ reflects the degree to which the underlying distribution is captured (Choi et al., 2022; Deja et al., 2023).

- *Representation learning through distribution approximation.* On the other hand, achieving high-quality distribution approximation results in more meaningful and informative representations in unsupervised learning. This is supported by Figure 2, where the findings, inspired by recent works (Zhang et al., 2023), demonstrate that diffusion models transition from memorizing the training data distribution to accurately approximating the underlying data distribution as the amount of training data increases. Consequently, better approximation of the underlying data distribution improves the quality of representation learning.

Given this relationship, we use posterior estimation as a proxy for representation quality throughout our analysis. Additionally, since diffusion models tend to memorize the training data instead of learning underlying data distribution when the training dataset is small (Zhang et al., 2023), we focus on the case where sufficient training data is available throughout our analysis in Section 3.

**Unimodal curve of representation quality.** Previous studies (Xiang et al., 2023; Baranchuk et al., 2021; Tang et al., 2023) have empirically shown that the representation dynamics of diffusion models follow a unimodal curve as the noise scale increases, across various tasks such as classification, segmentation, and image correspondence. Our findings corroborate this observation, as demonstrated in Figure 1(a), where the representation quality consistently exhibits a unimodal trend, regardless of the specific network architecture or dataset used (see Figure 1(c)-(d)). In the following analysis, we argue that this unimodal behavior arises from subtle differences between the requirements of representation learning and the generative nature of diffusion models.

High-fidelity image generation demands that diffusion models capture every aspect of the data distribution—from coarse structures to fine details. In contrast, representation learning, particularly for high-level tasks such as classification (Allen-Zhu & Li, 2022), prefers an abstract representation, where finer image details may even act as 'noise' that hinders performance. As shown in Figure 1(b), as the noise level increases, the predicted posteriors for clean input $\boldsymbol{x}_0$ transition from 'fine' to 'coarse' (Wang & Vastola, 2023; Choi et al., 2022), gradually removing fine-grained details. For the classification task in the plot, the best performance is achieved when the posterior estimation retains the essential information while discarding some class-irrelevant details. These findings indicate a trade-off between generative quality and representation performance (Chen et al., 2024b), prompting us to attribute variations in feature quality across noise levels to differences in posterior prediction.

## 3 THEORETICAL UNDERSTANDING THROUGH LOW-DIMENSIONAL MODELS

In this section, we theoretically examine the representation learning capabilities of diffusion models across varying noise levels by evaluating the quality of posterior estimation, $\mathbb{E}[\boldsymbol{x}_0|\boldsymbol{x}_t]$ for low-dimensional distributions.

### 3.1 ASSUMPTIONS OF LOW-DIMENSIONAL DATA DISTRIBUTION

Although real-world image datasets are high-dimensional in terms of pixel count and data volume, extensive empirical studies Gong et al. (2019); Pope et al. (2021); Stanczuk et al. (2022) suggest that their intrinsic dimensionality is considerably lower. Moreover, state-of-the-art large-scale diffusion models (Peebles & Xie, 2023; Podell et al., 2023) commonly employ auto-encoders (Kingma, 2013) to map images to a low-dimensional latent space (Rombach et al., 2022) for better training efficiency. Consequently, image datasets often reside on a union of low-dimensional manifolds.

In light of this, many recent studies of diffusion models have been focused on approximating low-dimensional distributions (Wang et al., 2024). Moreover, as union of low-dimensional manifolds can be locally approximated by a union of linear subspaces, it motivates us to model the underlying data distribution as a mixture of low-rank Gaussians (MoLRG) (Wang et al., 2024). The data points generated by MoLRG lie on a union of subspaces. Within each subspace, the data follows a Gaussian distribution with a low-rank covariance matrix that represents the subspace basis. Formally, we introduce a noisy version of the MoLRG distribution as follows:

**Assumption 1** ($K$-Subspace Noisy MoLRG Distribution). *For any sample $x_0$ drawn from the noisy MoLRG distribution with $K$ subspaces, the following holds:*

$$x_0 = U_k a + \delta U_k^\perp e, \text{ with probability } \pi_k \geq 0, \ k \in [K]. \tag{4}$$

*Here, $\sum_{k=1}^K \pi_k = 1$, $U_k \in \mathcal{O}^{n \times d_k}$ denotes an orthonormal basis for the $k$-th subspace, $d_k$ is the subspace dimension with $d_k \ll n$, and the coefficient $a \overset{i.i.d.}{\sim} \mathcal{N}(0, I_{d_k})$ is drawn from a standard normal distribution. For the noise, we assume $e \overset{i.i.d.}{\sim} \mathcal{N}(0, I_{n-d_k})$ with magnitude controlled by the scalar $\delta < 1$. Additionally, $U_k^\perp \in \mathcal{O}^{n \times (n-d_k)}$ is the orthogonal compliment of $U_k$.*

For simplicity of analysis, we let $d_1 = \cdots = d_K = d$, and we assume that the basis $\{U_k\}$ are orthogonal to each other with $U_k^T U_l = 0$ for all $k \neq l$. Additionally, we assume all mixing weights $\{\pi_k\}$ are equal with $\pi_1 = \cdots = \pi_K = 1/K$, and we define $U_\perp = \bigcap_{k=1}^K U_k^\perp \in \mathcal{O}^{n \times (n-Kd)}$ to be the noise space that is the orthogonal complement to all basis $\{U_k\}_{k=1}^K$.

We note that the noise term $\delta U_k^\perp e_i$ captures perturbations unrelated to the $k$-th subspace via the orthogonal complement $U_k^\perp$, thereby aligning the model more closely with real-world scenarios. These perturbations can be interpreted as attributes irrelevant to the subspace, such as the background in an image of a bird or the color/texture of a car. The extra noise term may not be relevant for representation learning, but it plays an importance role for diffusion model to generate high-fidelity samples. Additionally, for the noisy MoLRG distribution, ground truth posterior mean $\mathbb{E}[\hat{x}_0|x_t]$ is:

**Proposition 1.** *For a $K$-class MoLRG data distribution, for each time $t > 0$, it holds that*

$$\hat{x}_\theta^\star(x_t, t) := \mathbb{E}[\hat{x}_0|x_t] = \sum_{k=1}^K w_k^\star(x_t) \left( \frac{1}{1+\sigma_t^2} U_k U_k^T + \frac{\delta^2}{\delta^2 + \sigma_t^2} U_k^\perp U_k^{\perp T} \right) x_t \tag{5}$$

$$\text{where } w_k^\star(x_t) := \frac{\exp(g_k(x_t, t))}{\sum_{k=1}^K \exp(g_k(x_t, t))}, \tag{6}$$

$$\text{and } g_k(x) = \frac{1}{2\sigma_t^2(1+\sigma_t^2)} \|U_k^T x\|^2 + \frac{\delta^2}{2\sigma_t^2(\delta^2+\sigma_t^2)} \|U_k^{\perp T} x\|^2. \tag{7}$$

**Remark.** In the above proposition, we present the ground truth posterior estimation function that a diffusion model can achieve by minimizing the training objective defined in (3). We denote this optimal model $\hat{x}_\theta^\star$. Given the established relationship between posterior estimation and representation learning on clean inputs $x_0$, we can now analyze the representation learning dynamics under this optimal setting by evaluating $\hat{x}_\theta^\star(x_0, t)$ at different time step $t$.

### 3.2 MAIN THEORETICAL RESULTS

As we discussed in Section 2.3, based upon the strong correlation between representation quality and the posterior mean estimation, we analyze $\hat{x}_\theta^\star(x_0, t)$ across different time step $t \in [0, 1]$. Here, we use $x_0$ as the input instead of $x_t$ according to our discussion in Section 2.2.

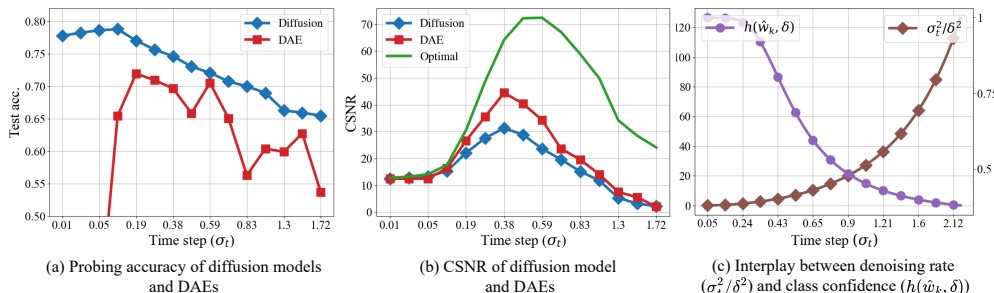

Figure 4: **Dynamics of feature probing accuracy, CSNR, and denoising/class confidence rate with increasing noise levels.** Panels (a) and (b) show the feature probing accuracy and CSNR trends using the same MoLRG data as in Figure 3, both exhibiting a unimodal pattern. The interplay between the "denoising rate" and the class confidence rate for the approximate optimal solution $\hat{x}_{approx}^{\star}$ is illustrated in panel (c).

Given $x_0 \sim$ MoLRG and without loss of generality, let $k$ represent the true class to which $x_0$ belongs. We quantify the accuracy of posterior mean estimation by introducing a measure of Class-specific Signal-to-Noise Ratio (CSNR) as follows:

$$\mathrm{CSNR}(t, \hat{x}_{\boldsymbol{\theta}}^{\star}) := \frac{\mathbb{E}_{x_0}[\|U_k U_k^T \hat{x}_{\boldsymbol{\theta}}^{\star}(x_0, t)\|^2]}{\mathbb{E}_{x_0}[\sum_{l \neq k} \|U_l U_l^T \hat{x}_{\boldsymbol{\theta}}^{\star}(x_0, t)\|^2]} \tag{8}$$

We know that successful prediction of the class for $x_0$ occurs when the class-specific signal $\|U_k U_k^T \hat{x}_{\boldsymbol{\theta}}^{\star}(x_0, t)\|$ dominates over the noise term $\|U_k^{\perp} U_k^{\perp T} \hat{x}_{\boldsymbol{\theta}}^{\star}(x_0, t)\|$. On the other hand, because

$$\|U_k^{\perp} U_k^{\perp T} \hat{x}_{\boldsymbol{\theta}}^{\star}(x_0, t)\|^2 = \sum_{l \neq k} \|U_l U_l^T \hat{x}_{\boldsymbol{\theta}}^{\star}(x_0, t)\|^2 + \|U_{\perp} U_{\perp}^T \hat{x}_{\boldsymbol{\theta}}^{\star}(x_0, t)\|^2$$

and $U_{\perp}$ does not affect classification due to its presence in every data point, it leads to our definition of CSNR in equation 8 which measures the ratio between the true class signal and irrelevant noise from other classes at a given noise level for a specific posterior estimation function. We note that CSNR is defined with respect to two variables: the timestep $t$ and a posterior predicting function $f$. Therefore, it can be evaluated for any specified posterior prediction function at a given timestep.

Therefore, intuitively, a higher CSNR indicates a better recovery of the underlying low-dimensional data subspace, and thus the predicted posterior is more likely to be assigned to the correct class. This is supported by Figure 4(a)-(b) which shows that both CSNR$(t)$ and classification accuracy using the learned representation follow similar unimodal curves.

To simplify the calculation of (8), which involves the expectation over the softmax term $w_k^{\star}$, we approximate $\hat{x}_{\boldsymbol{\theta}}^{\star}$ as follows:

$$\hat{x}_{approx}^{\star}(x, t) = \sum_{k=1}^{K} \hat{w}_k \left( \frac{1}{1 + \sigma_t^2} U_k U_k^T + \frac{\delta^2}{\delta^2 + \sigma_t^2} U_k^{\perp} U_k^{\perp T} \right) x,$$

$$\text{where } \hat{w}_k := \frac{\exp\left(\mathbb{E}_{x_0}[g_k(x_0, t)]\right)}{\sum_{k=1}^{K} \exp\left(\mathbb{E}_{x_0}[g_k(x_0, t)]\right)}. \tag{9}$$

In other words, we use $\hat{w}_k$ in equation 9 to approximate $w_k^{\star}(x_0)$ in equation 6 by taking expectation inside the softmax with respect to $x_0$. This allows us to treat $\hat{w}_k$ as a constant when calculating CSNR, making the analysis more tractable while maintaining $\mathbb{E}[\|U_l U_l^T \hat{x}_{\boldsymbol{\theta}}^{\star}(x_0, t)\|^2] \approx \mathbb{E}[\|U_l U_l^T \hat{x}_{approx}^{\star}(x, t)(x_0, t)\|^2]$ for all $l \in [K]$. We verify the tightness of this approximation at Appendix A.3 (Figure 9). Now, we are ready to state our main theorem as follows.

**Theorem 1.** *Let data $x_0$ be any arbitrary data point drawn from the MoLRG distribution defined in Assumption 1 and let $k$ denote the true class $x_0$ belongs to. Then CSNR introduced in equation 8 depends on the noise level $\sigma_t$ in the following form:*

$$\mathrm{CSNR}(t, \hat{x}_{approx}^{\star}) = \frac{1}{(K-1)\delta^2} \cdot \left( \frac{1 + \frac{\sigma_t^2}{\delta^2} h(\hat{w}_k, \delta)}{1 + \frac{\sigma_t^2}{\delta^2} h(\hat{w}_l, \delta)} \right)^2 \tag{10}$$

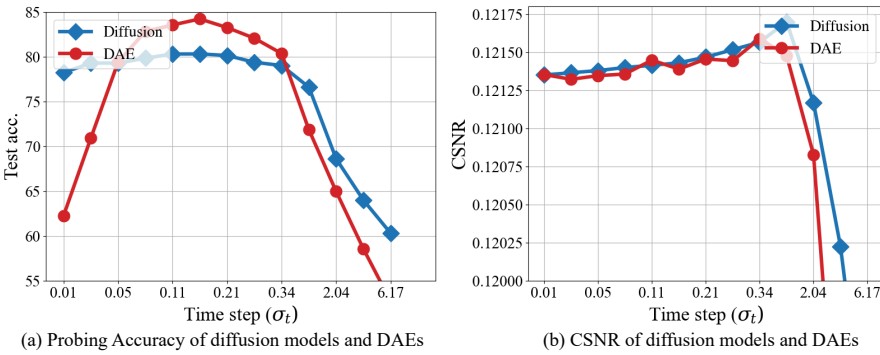

Figure 5: **Dynamics of feature probing accuracy and** CSNR **on CIFAR10.** Panels (a) and (b) show the feature probing accuracy and CSNR trends computed using the CIFAR10 test dataset, both exhibiting a unimodal pattern.

*where $h(w, \delta) := (1 - \delta^2)w + \delta^2$. Since $\delta$ is fixed, $h(w, \delta)$ is a monotonically increasing function with respect to $w$. Note that here $\delta$ represents the magnitude of the fixed intrinsic noise in the data where $\sigma_t$ denotes the level of additive Gaussian noise introduced during the diffusion training process.*

**Remark.** Intuitively, the unimodal curve of CSNR reflects how the additive noise level $\sigma_t$ in the diffusion process helps counteract the intrinsic data noise $\delta$. The noise ratio $(\sigma_t/\delta)$ can be interpreted as the "denoising" rate, where a larger ratio indicates more data noise being canceled out and vice versa. Meanwhile, $h(\hat{w}_k, \delta)$ represents the class confidence rate, with lower values meaning less class-specific information is captured by the model. With $\sigma_t$ increases from 0 to $\infty$, the "denoising rate" rises accordingly, while the class confidence rate decreases monotonically. Thus, from Theorem 1, we can derive the rationale behind the unimodal behavior of CSNR.

- **The unimodal curve of** CSNR**.** The unimodal curve is decided by the interplay between the "denoising rate" and the class confidence rate as noise increases. As observed in Figure 4(c), the "denoising rate" $(\sigma_t^2/\delta^2)$ increases monotonically with $\sigma_t$ while the class confidence rate $h(\hat{w}_k, \delta)$ monotonically declines. Initially, as $\sigma_t$ increases, the class confidence rate remains relatively stable due to its flat slope (as seen in Figure 4(c)), and an increasing "denoising rate" enhances the CSNR, resulting in improved posterior estimation. However, as indicated by (7), when $\sigma_t$ becomes too large, $h(\hat{w}_k, \delta)$ approaches $h(\hat{w}_l, \delta)$, leading to a drop in CSNR, which limits the model's ability to project $\boldsymbol{x}_0$ onto the correct signal space and ultimately impairs posterior estimation. This interpretation is validated by the visualization in Figure 3. In the plot, each class is represented by a colored straight line, while deviations from these lines correspond to the $\delta$-related noise term. Initially, increasing the noise scale effectively cancels out the $\delta$-related data noise, resulting in a cleaner posterior estimation and improved probing accuracy. However, as the noise continues to increase, the class confidence rate drops, leading to an overlap between classes, which ultimately degrades the feature quality and probing performance.

Back to our real-world analogy, the proportion of data associated with $\delta$ represents class-irrelevant attributes or finer image details. The unimodal representation learning dynamic thus captures a "fine-to-coarse" shift (Choi et al., 2022; Wang & Vastola, 2023), where these details are progressively stripped away. During this process, peak representation performance is achieved at a balance point where class-irrelevant attributes are eliminated, while class-essential information is preserved.

### 3.3 EMPIRICAL VALIDATION

In this subsection, we conduct experiments on both synthetic and real datasets to validate our theory on the representation learning dynamics.

We use two datasets: a 3-class MoLRG dataset, where each subspace has dimension $d = 1$ and ambient dimension $n = 10$, with noise scale $\delta = 0.2$ , and the standard CIFAR10 dataset (Krizhevsky et al., 2009). We consider two training settings: (a) a DDPM-based diffusion training configuration and (b) a vanilla DAE training configuration, where separate DAEs are trained for different noise levels. Here, the separate DAEs serve as a control group, enabling us to isolate the effects of the denoising process, as discussed in Section 2.1. We leave further training details in Appendix A.2.

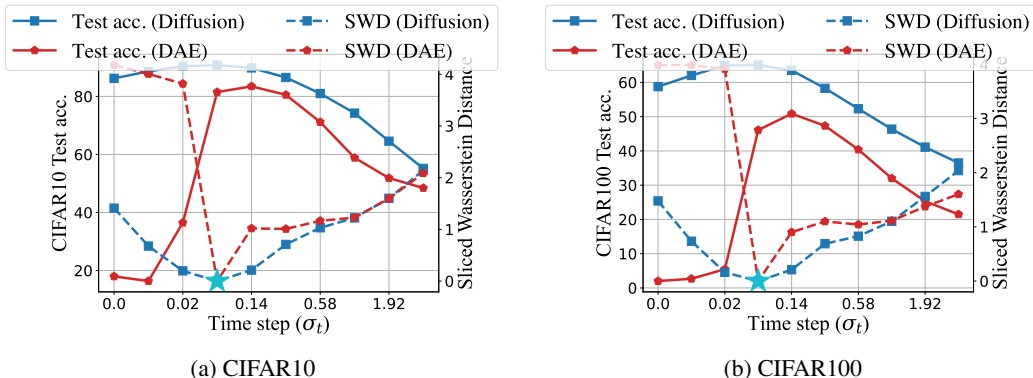

(a) CIFAR10              (b) CIFAR100

Figure 6: **Comparison of representation learning performance and feature similarity between diffusion model and individual DAEs.** We train DDPM-based diffusion models and individual DAEs on the CIFAR10 and CIFAR100 datasets. After training, we plotted their representation learning performance and feature similarity against the best features (indicated by ⋆) as the noise level increases.

After training, we extract intermediate features and posterior predictions from both diffusion models and DAEs, followed by linear probing on the features and computation of empirical CSNR for the posterior estimations. The results for the two datasets are presented in Figure 4 and Figure 5, respectively. As shown in the plots, both feature probing accuracy and the empirical CSNR exhibit a matching unimodal curve, consistent across training configurations and datasets, thus supporting our theoretical results.

## 4 ADDITIONAL EXPERIMENTS

In Section 3, we analyzed diffusion representation dynamics with a focus on the denoising process, assuming sufficient training data for learning the underlying distribution. In this section, we explore the impact of the diffusion process (Section 4.1) and data complexity (Section 4.2) in shaping diffusion models' representation learning dynamics.

### 4.1 WEIGHT SHARING IN DIFFUSION MODELS HELPS REPRESENTATION LEARNING

In this subsection, we demonstrate how the inherent weight-sharing mechanism in diffusion models, stemming from their loss design, enhances representation learning performances compared with traditional DAEs.

Previously, in Section 3, we analyzed the optimal posterior function by treating each noise level independently. However, the training objective for diffusion models in (3) involves minimizing the loss across all noise levels simultaneously, which results in interactions and parameter sharing among denoising subcomponents at different noise levels. We hypothesize that these interactions and parameter sharing create greater feature similarity across noise scales, effectively functioning as an implicit "ensemble" mechanism that enhances the performance of diffusion models compared to individual DAEs (Chen et al., 2024b), which accounts for the significant performance gap between DAEs and diffusion models, as shown in Figure 4(a) and Figure 5(a).

To test this hypothesis, we trained 10 individual DAEs, each at a different noise level, as well as a single DDPM-based diffusion model on CIFAR10 and CIFAR100 datasets. We then conducted linear probing on the features extracted from both setups. To evaluate feature similarity, we calculated the sliced Wasserstein distance (SWD) (Doan et al., 2024) between features for both diffusion and DAE models at various noise levels and their corresponding features at $\sigma_t = 0.06$, which achieves near-optimal accuracy for all scenarios.

As shown in Figure 6, diffusion models consistently outperform individual DAEs, particularly at lower noise levels, where the performance gap is most pronounced. In these low-noise regions, due to the almost negligible additive noise, individual DAEs are more likely to be trained as identity functions, leading to trivial representations. In contrast, the parameter sharing in diffusion models alleviates this issue significantly. The SWD curve demonstrates an inverse correlation with the test

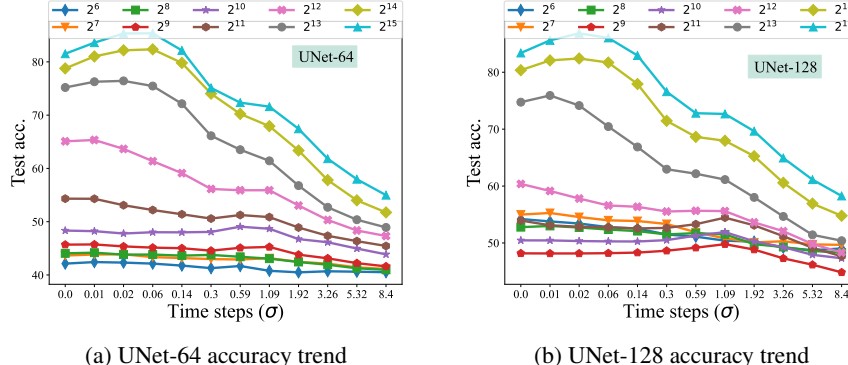

(a) UNet-64 accuracy trend        (b) UNet-128 accuracy trend

Figure 7: **The influence of data complexity in diffusion-based representation learning.** With the same model trained in Figure 2, we plot the representation learning dynamics for each trained model as a function of changing noise levels.

accuracy curve, indicating that features closer to their optimal state possess stronger representational capacity. Furthermore, the plot shows that diffusion model features across different noise levels remain significantly closer to their optimal features at $\sigma_t = 0.06$, while DAE features show less similarity. These results strongly support our hypothesis.

The concept of this "sharing mechanism" is also supprted by previous empirical studies on DAEs, which have shown that sequential training over multiple noise scales enhances representation quality (Chandra & Sharma, 2014; Geras & Sutton, 2014; Zhang & Zhang, 2018). In this work, we conduct an ablation study to explore methods for improving DAE performance at lower noise levels, finding that training with multiple noise scales provided the most promising results. Further details can be found in Appendix A.3 (Table 1).

### 4.2   THE INFLUENCE OF DATA COMPLEXITY IN DIFFUSION REPRESENTATION LEARNING

So far, our analyses are based on the assumption that the training dataset contains sufficient samples for the diffusion model to learn the underlying distribution. Interestingly, if this assumption is violated by training the model on insufficient data, the unimodal representation learning dynamic disappears and the probing accuracy also drops severely.

As illustrated in Figure 7, we train 2 different UNets following the EDM (Karras et al., 2022) configuration with training dataset size ranging from $2^5$ to $2^{15}$. The unimodal curve emerges only when the dataset size exceeds $2^{12}$, whereas smaller datasets produce flat curves.

The underlying reason for this observation is that, when training data is limited, diffusion models memorize all individual data points rather than learn the true underlying data structure (Wang et al., 2024). In this scenario, the model memorizes an empirical distribution that lacks meaningful low-dimensional structures and thus deviates from the setting in our theory, leading to the loss of the unimodal representation dynamic. To confirm this, we calculated the generalization score, which measures the percentage of generated data that does not belong to the training dataset, as defined in (Zhang et al., 2023). As shown in Figure 2, representation learning only achieves strong accuracy and displays the unimodal dynamic when the generalization score approaches 1, aligning with our theoretical assumptions.

## 5   CONCLUSION

In this work, we establish a link between distribution recovery, posterior estimation, and representation learning, providing the first theoretical study of diffusion-based representation learning dynamics across varying noise scales. Using a low-dimensional mixture of low-rank Gaussians, we show that the unimodal representation learning dynamic arises from the interplay between data denoising and class specification. Additionally, our analysis highlights the inherent weight-sharing mechanism in diffusion models, demonstrating its benefits for peak representation performance as well as its limitations in optimizing high-noise regions due to increased complexity. Experiments on both synthetic and real datasets validate our findings.

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

# A APPENDIX

The Appendis is organized as follows: in Appendix A.1, we discuss related works; in Appendix A.2, we present the detailed experimental setups for the empirical results in the paper; in Appendix A.3, we provide complementary experiments. Lastly, in Appendix A.4, we provide proof details for Section 3.

## A.1 RELATED WORKS

**Denoising auto-encoders.** Denoising autoencoders (DAEs) are trained to reconstruct corrupted images to extract semantically meaningful information, which can be applied to various vision (Vincent et al., 2008; 2010) and language downstream tasks (Lewis, 2019). Related to our analysis of the weight-sharing mechanism, several studies have shown that training with a noise scheduler can enhance downstream performance (Chandra & Sharma, 2014; Geras & Sutton, 2014; Zhang & Zhang, 2018). On the theoretical side, prior works have studied the learning dynamics (Pretorius et al., 2018; Steck, 2020) and optimization landscape (Kunin et al., 2019) through the simplified linear DAE models.

**Diffusion-based representation learning.** Diffusion-based representation learning Fuest et al. (2024) has demonstrated significant success in various downstream tasks, including image classification (Xiang et al., 2023; Mukhopadhyay et al., 2023; Deja et al., 2023), segmentation (Baranchuk et al., 2021), correspondence (Tang et al., 2023), and image editing (Shi et al., 2024). To further enhance the utility of diffusion features, knowledge distillation (Yang & Wang, 2023; Li et al., 2023) methods have been proposed, aiming to bypass the computationally expensive grid search for the optimal $t$ in feature extraction and improving downstream performance. Beyond directly using intermediate features from pre-trained diffusion models, research efforts has also explored novel loss functions (Abstreiter et al., 2021; Wang et al., 2023) and network modifications (Hudson et al., 2024; Preechakul et al., 2022) to develop more unified generative and representation learning capabilities within diffusion models. Unlike the aforementioned efforts, our work focuses more on understanding the representation learning capabilities of diffusion models.

## A.2 EXPERIMENTAL DETAILS

In this section, we provide technical details for all the experiments in the main body of the paper.

**Experimental details for Figure 1 (a)-(b).** We utilize a minimal implementation of the original DDPM model from an online public repository (BohaoZou, 2022), consisting of a 12-layer UNet (including input/output embedding layers), and train it on the CIFAR10 dataset with $T = 1000$ time steps for 200 epochs with an AdamW optimizer and learning rate $1 \times 10^{-4}$. Features are extracted as 512-dimensional vectors from the output of the 7th layer (i.e., the bottleneck layer) at time steps [1, 5, 10, 20, 30, 40, 60, 80, 100, 200, 400, 500, 600], each corresponding to a specific $\sigma_t$ ranging from 0.01 to 6.17. Linear probing is applied to the extracted features, as in (Xiang et al., 2023), to plot the feature probing accuracy curve in Figure 1(a). For the posterior estimation ($x_\theta(x_0, t)$) probing accuracy curve, also shown in Figure 1(a), we use a two-layer MLP probe with ReLU activation. The estimated posterior at these time steps is visualized in Figure 1(b).

**Experimental details for Figure 1 (c)-(d).** We train diffusion models based on the unified framework proposed by Karras et al. (2022). Specifically, we use the DDPM+ network, and use EDM configuration for Figure 1 (c) while taking VP configuration Figure 1 (d). Karras et al. (2022) has shown equivalence between VP configuration and the traditional DDPM setting, thus we call the models in Figure 1 (d) as DDPM* models. For each of EDM and VP configuration, we train two models on CIFAR10 and CIFAR100, respectively. After training, we conduct linear probe on CIFAR10 and CIFAR100. At a specific noise level $\sigma(t)$, we either use clean image $x_0$ or noisy image $x_t = x_0 + n$ as input to the EDM or the DDPM* models for extracting features after the '8x8_block3' layer. Here, $n$ represents random noise and $n \sim \mathcal{N}(0, \sigma(t)^2 I)$. We train a logistic regression on features in the train split and report the classification accuracy on the test split of the dataset. We perform the linear probe for each of the following noise levels: [0.002, 0.008, 0.023, 0.060, 0.140, 0.296, 0.585, 1.088, 1.923, 3.257].

**Experimental details for Figure 3 and Figure 4.** For the MoLRG experiments, we train a 3-layer MLP with ReLU activation and a hidden dimension of 128, following the setup provided in an open-source repository (tanelp, 2022). The MLP is trained for 200 epochs using DDPM scheduling with $T = 500$, employing the Adam optimizer with a learning rate of $1 \times 10^{-3}$. For feature extraction, we use the activations of the second layer of the MLP (dimension 128) as intermediate features for linear probing. For CSNR computation, we follow the definition in Equation (8) since we have access to the ground-truth basis for the MoLRG data. In Figure 3, we visualize the posterior estimations at time steps [1, 20, 80, 200, 260] by projecting them onto the union of $U_1, U_2$, and $U_3$ (a 3D space), then further projecting onto the 2D plane along the (1, 1, 1) direction. The subtitles of each visualization show the corresponding probing accuracy and CSNR calculated as explained above. For Figure 4(a)(b), we plot the accuracy and CSNR at time steps [1, 5, 10, 20, 40, 60, 80, 100, 120, 140, 160, 180, 220, 240, 260]. We perform linear probing using the features extracted from the training set and test on five different MoLRG datasets generated with five different random seeds, reporting the average accuracy.

**Experimental details for Figure 5.** We use the same experimental settings as in Figure 1(a)(b). Additionally, we train individual DAEs for each different time step. The accuracy curves in Figure 5(a) are plotted identically as in Figure 1(a). The CSNR metric in Figure 5(b) is calculated from the definition Equation (8), with the basis $U_k$ for each CIFAR10 class estimated as the first five right singular vectors of the data from the $k$-th class.

**Experimental details for Figure 6.** We train individual DAEs using the DDPM++ network and VP configuration outlined in Karras et al. (2022) at the following noise scales: [0.002, 0.008, 0.023, 0.06, 0.14, 0.296, 0.585, 1.088, 1.923, 3.257]. Each model is trained for 500 epochs using the Adam optimizer (Kingma, 2014) with a fixed learning rate of $1 \times 10^{-4}$. For the diffusion models, we reuse the model from Figure 1(d). The sliced Wasserstein distance is computed according to the implementation described in Doan et al. (2024).

**Experimental details for Figure 7.** We use the DDPM++ network and VP configuration to train diffusion models(Karras et al., 2022) on the CIFAR10 dataset, using two network configurations: UNet-64 and UNet-128, by varying the embedding dimension of the UNet. Training dataset sizes range exponentially from $2^6$ to $2^{15}$. For each dataset size, both UNet-64 and UNet-128 are trained on the same subset of the training data. All models are trained with a duration of 50K images following the EDM training setup. After training, we calculate the generalization score as described in Zhang et al. (2023), using 10K generated images and the full training subset to compute the score.

### A.3 ADDITIONAL EXPERIMENTS

**Additional representation learning experiments on DDPM.** Apart from EDM and DDPM* models pre-trained using the framework proposed by Karras et al. (2022), we also experiment with the features extracted by classic DDPM models (Ho et al., 2020) to make sure the observations do not depend on the specific training framework. We use the same groups of noise levels and also test using clean or noisy images as input to extract features at the bottleneck layer, and then conduct the linear probe. The DDPM models we use are trained on the Flowers-102 (Nilsback & Zisserman, 2008) and the CIFAR10 dataset accordingly. Different from the framework proposed by Karras et al. (2022), the input to the classic DDPM model is the same as the input to the UNet inside. Therefore, we calculate the scaling factor $\sqrt{\bar{\alpha}_t} = 1/\sqrt{\sigma^2(t) + 1}$, and use $\sqrt{\bar{\alpha}_t} x_0$ as the clean image input. Besides, for noisy input, we set $x_t = \sqrt{\bar{\alpha}_t}(x_0 + n)$, with $n \sim \mathcal{N}(0, \sigma(t)^2 I)$. The linear probe results are presented in Figure 8, where we consistently see an unimodal curve, as well as compatible or even superior representation learning performance of clean input $x_0$.

**Validation of $\hat{x}^\star_{approx}$ approximation in Section 3.** In Section 3, we approximate the optimal posterior estimation function $\hat{x}^\star_{\boldsymbol{\theta}}$ using $\hat{x}^\star_{approx}$ by taking the expectation inside the softmax with respect to $x_0$. To validate this approximation, we compare the CSNR calculated from $\hat{x}^\star_{\boldsymbol{\theta}}$ and from $\hat{x}^\star_{approx}$ using (8) and (9), respectively. We use a fixed dataset size of 2400 and set the default parameters to $n = 50$, $d = 5$, $K = 3$, and $\delta = 0.1$ to generate MoLRG data. We then vary one parameter at a time while keeping the others constant, and present the computed CSNR in Figure 9. As shown, the approximated CSNR score consistently aligns with the actual score.

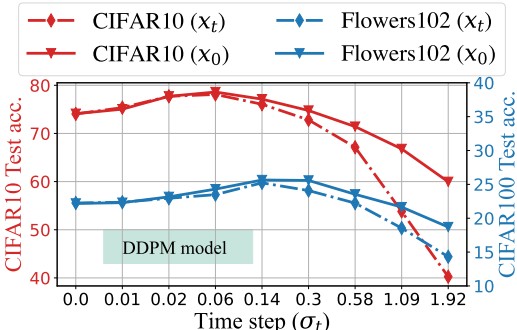

Figure 8: **Performance comparison: clean vs. noisy inputs.** We use pre-trained DDPM model on the Flowers-102 (Nilsback & Zisserman, 2008) and CIFAR10 dataset. The feature probing accuracy is plotted to compare the performance when using clean versus noisy inputs.

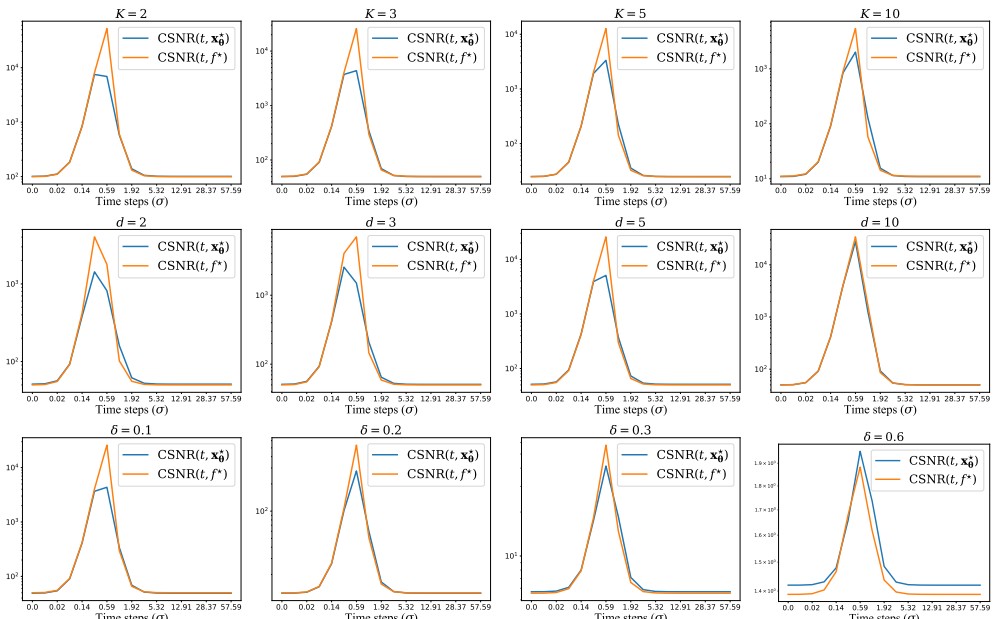

Figure 9: **Comparison between** CSNR **calculated using the optimal model** $\hat{x}_\theta^\star$ **and the** CSNR **calculated with our approximation in Theorem 1.** We generate `MoLRG` data and calculate CSNR using both the corresponding optimal posterior function $\hat{x}_\theta^\star$ and our approximation $\hat{x}_{approx}^\star$ from Theorem 1. Default parameters are set as $n = 50$, $d = 5$, $K = 3$, and $\delta = 0.1$. In each row, we vary one parameter while keeping the others fixed, comparing the actual and approximated CSNR.

**Mitigating the performance gap between DAE and diffusion models.** Throughout the empirical results presented in this paper, we consistently observe a performance gap between individual DAEs and diffusion models, especially in low-noise regions. Here, we use a DAE trained on the CIFAR-10 dataset with a single noise level $\sigma = 0.002$, using the NCSN++ architecture (Karras et al., 2022). In the default setting, the DAE achieves a test accuracy of 32.3. We then explore three methods to improve the test performance: (a) adding dropout, as noise regularization and dropout have been effective in preventing autoencoders from learning identity functions (Steck, 2020); (b) adopting EDM-based preconditioning during training, including input/output scaling, loss weighting, etc.; and (c) multi-level noise training, in which the DAE is trained simultaneously on three noise levels $[0.002, 0.012, 0.102]$. Each modification is applied independently, and the results are reported in Table 1. As shown, dropout helps improve performance, but even with a dropout rate of $0.95$, the improvement is minor. EDM-based preconditioning achieves moderate improvement, while multi-

Table 1: **Improve DAE representation performance at low noise region.** A vanilla DAE trained on the CIFAR-10 dataset with a single noise level of $\sigma = 0.002$ serves as the baseline. We evaluate the performance improvement of dropout regularization, EDM-based preconditioning, and multi-level noise training ($\sigma = \{0.002, 0.012, 0.102\}$). Each technique is applied independently to assess its contribution to performance enhancement.

| Modifications | Test acc. |
|---|---|
| Vanilla DAE | 32.3 |
| +Dropout (0.5) | 35.3 |
| +Dropout (0.9) | 36.4 |
| +Dropout (0.95) | 38.1 |
| +EDM preconditioning | 49.2 |
| +Multi-level noise training | **58.6** |

level noise training yields the most promising results, demonstrating the benefit of incorporating the diffusion process in DAE training.

### A.4 PROOFS

#### A.4.1 PROOF OF PROPOSITION 1

*Proof.* We follow the same proof steps as in (Wang et al., 2024) Lemma 1 with a change of variable. Let $\boldsymbol{c}_k = \begin{bmatrix} \boldsymbol{a}_k \\ \boldsymbol{e}_k \end{bmatrix}$ and $\widetilde{\boldsymbol{U}_k} = \begin{bmatrix} \boldsymbol{U}_k & \delta \boldsymbol{U}_k^\perp \end{bmatrix}$, we first compute

$$p_t(\boldsymbol{x}|Y=k)$$

$$= \int p_t(\boldsymbol{x}|Y=k, \boldsymbol{c}_k) \mathcal{N}(\boldsymbol{c}_k; \boldsymbol{0}, \boldsymbol{I}_{d+D}) \, \mathrm{d}\boldsymbol{c}_k$$

$$= \int p_t(\boldsymbol{x}|\boldsymbol{x}_0 = \widetilde{\boldsymbol{U}_k}\boldsymbol{c}_k) \mathcal{N}(\boldsymbol{c}_k; \boldsymbol{0}, \boldsymbol{I}_{d+D}) \, \mathrm{d}\boldsymbol{c}_k$$

$$= \int \mathcal{N}(\boldsymbol{x}; s_t \widetilde{\boldsymbol{U}_k}\boldsymbol{c}_k, \gamma_t^2 \boldsymbol{I}_n) \mathcal{N}(\boldsymbol{c}_k; \boldsymbol{0}, \boldsymbol{I}_{d+D}) \, \mathrm{d}\boldsymbol{c}_k$$

$$= \frac{1}{(2\pi)^{n/2}(2\pi)^{(d+D)/2}\gamma_t^n} \int \exp\left(-\frac{1}{2\gamma_t^2}\|\boldsymbol{x} - s_t\widetilde{\boldsymbol{U}_k}\boldsymbol{c}_k\|^2\right) \exp\left(-\frac{1}{2}\|\boldsymbol{c}_k\|^2\right) \mathrm{d}\boldsymbol{c}_k$$

$$= \frac{1}{(2\pi)^{n/2}(2\pi)^{(d+D)/2}\gamma_t^n} \int \exp\left(-\frac{1}{2\gamma_t^2}\left(\boldsymbol{x}^T\boldsymbol{x} - 2s_t\boldsymbol{x}^T\widetilde{\boldsymbol{U}_k}\boldsymbol{c}_k + s_t^2\boldsymbol{c}_k^T\widetilde{\boldsymbol{U}_k}^T\widetilde{\boldsymbol{U}_k}\boldsymbol{c}_k + \gamma_t^2\boldsymbol{c}_k^T\boldsymbol{c}_k\right)\right) \mathrm{d}\boldsymbol{c}_k$$

$$= \frac{1}{(2\pi)^{n/2}\gamma_t^n} \left(\frac{s_t^2 + \gamma_t^2}{\gamma_t^2}\right)^{-d/2} \left(\frac{s_t^2\delta^2 + \gamma_t^2}{\gamma_t^2}\right)^{-D/2} \exp\left(-\frac{1}{2\gamma_t^2}\boldsymbol{x}^T\left(\boldsymbol{I}_n - \frac{s_t^2}{s_t^2 + \gamma_t^2}\boldsymbol{U}_k\boldsymbol{U}_k^T - \frac{s_t^2\delta^2}{s_t^2\delta^2 + \gamma_t^2}\boldsymbol{U}_k^\perp\boldsymbol{U}_k^{\perp T}\right)\boldsymbol{x}\right)$$

$$\int \frac{1}{(2\pi)^{d/2}} \left(\frac{\gamma_t^2}{s_t^2 + \gamma_t^2}\right)^{-d/2} \exp\left(-\frac{s_t^2 + \gamma_t^2}{2\gamma_t^2}\left\|\boldsymbol{a}_k - \frac{s_t}{s_t^2 + \gamma_t^2}\boldsymbol{U}_k^T\boldsymbol{x}\right\|^2\right) \mathrm{d}\boldsymbol{a}_k$$

$$\int \frac{1}{(2\pi)^{D/2}} \left(\frac{\gamma_t^2}{s_t^2\delta^2 + \gamma_t^2}\right)^{-D/2} \exp\left(-\frac{s_t^2\delta^2 + \gamma_t^2}{2\gamma_t^2}\left\|\boldsymbol{e}_k - \frac{s_t\delta}{s_t^2\delta^2 + \gamma_t^2}\boldsymbol{U}_k^{\perp T}\boldsymbol{x}\right\|^2\right) \mathrm{d}\boldsymbol{e}_k$$

$$= \frac{1}{(2\pi)^{n/2}} \frac{1}{(s_t^2 + \gamma_t^2)^{d/2}(s_t^2\delta^2 + \gamma_t^2)^{D/2}} \exp\left(-\frac{1}{2\gamma_t^2}\boldsymbol{x}^T\left(\boldsymbol{I}_n - \frac{s_t^2}{s_t^2 + \gamma_t^2}\boldsymbol{U}_k\boldsymbol{U}_k^T - \frac{s_t^2\delta^2}{s_t^2\delta^2 + \gamma_t^2}\boldsymbol{U}_k^\perp\boldsymbol{U}_k^{\perp T}\right)\boldsymbol{x}\right)$$

$$= \frac{1}{(2\pi)^{n/2} \det^{1/2}(s_t^2\boldsymbol{U}_k\boldsymbol{U}_k^T + s_t^2\delta^2\boldsymbol{U}_k^\perp\boldsymbol{U}_k^{\perp T} + \gamma_t^2\boldsymbol{I}_n)}$$

$$\exp\left(-\frac{1}{2}\boldsymbol{x}^T\left(s_t^2\boldsymbol{U}_k\boldsymbol{U}_k^T + s_t^2\delta^2\boldsymbol{U}_k^\perp\boldsymbol{U}_k^{\perp T} + \gamma_t^2\boldsymbol{I}_n\right)^{-1}\boldsymbol{x}\right)$$

$$= \mathcal{N}(\boldsymbol{x}; \boldsymbol{0}, s_t^2\boldsymbol{U}_k\boldsymbol{U}_k^T + s_t^2\delta^2\boldsymbol{U}_k^\perp\boldsymbol{U}_k^{\perp T} + \gamma_t^2\boldsymbol{I}_n),$$

where we repeatedly apply the pdf of multi-variate Gaussian and the second last equality uses $\det(s_t^2 \boldsymbol{U}_k \boldsymbol{U}_k^T + s_t^2 \delta^2 \boldsymbol{U}_k^\perp \boldsymbol{U}_k^{\perp T} + \gamma_t^2 \boldsymbol{I}_n) = (s_t^2 + \gamma_t^2)^d (s_t^2 \delta^2 + \gamma_t^2)^D$ and $(s_t^2 \boldsymbol{U}_k \boldsymbol{U}_k^T + s_t^2 \delta^2 \boldsymbol{U}_k^\perp \boldsymbol{U}_k^{\perp T} + \gamma_t^2 \boldsymbol{I}_n)^{-1} = \left(\boldsymbol{I}_n - s_t^2/(s_t^2 + \gamma_t^2) \boldsymbol{U}_k \boldsymbol{U}_k^T - s_t^2 \delta^2/(s_t^2 \delta^2 + \gamma_t^2) \boldsymbol{U}_k^\perp \boldsymbol{U}_k^{\perp T}\right)/\gamma_t^2$ because of the Woodbury matrix inversion lemma. Hence, with $\mathbb{P}(Y = k) = \pi_k$ for each $k \in [K]$, we have

$$p_t(\boldsymbol{x}) = \sum_{k=1}^K p_t(\boldsymbol{x}|Y = k)\mathbb{P}(Y = k) = \sum_{k=1}^K \pi_k \mathcal{N}(\boldsymbol{x}; \boldsymbol{0}, s_t^2 \boldsymbol{U}_k \boldsymbol{U}_k^T + s_t^2 \delta^2 \boldsymbol{U}_k^\perp \boldsymbol{U}_k^{\perp T} + \gamma_t^2 \boldsymbol{I}_n).$$

Now we can compute the score function

$$
\begin{aligned}
\nabla \log p_t(\boldsymbol{x}) = \frac{\nabla p_t(\boldsymbol{x})}{p_t(\boldsymbol{x})} &= \frac{\sum_{k=1}^K \pi_k \mathcal{N}(\boldsymbol{x}; \boldsymbol{0}, s_t^2 \boldsymbol{U}_k \boldsymbol{U}_k^T + s_t^2 \delta^2 \boldsymbol{U}_k^\perp \boldsymbol{U}_k^{\perp T} + \gamma_t^2 \boldsymbol{I}_n)}{\sum_{k=1}^K \pi_k \mathcal{N}(\boldsymbol{x}; \boldsymbol{0}, s_t^2 \boldsymbol{U}_k \boldsymbol{U}_k^T + s_t^2 \delta^2 \boldsymbol{U}_k^\perp \boldsymbol{U}_k^{\perp T} + \gamma_t^2 \boldsymbol{I}_n)} \\
&\quad \cdot \frac{\left(-\frac{1}{\gamma_t^2}\boldsymbol{x} + \frac{s_t^2}{\gamma_t^2(s_t^2 + \gamma_t^2)}\boldsymbol{U}_k \boldsymbol{U}_k^T \boldsymbol{x} + \frac{s_t^2 \delta^2}{\gamma_t^2(s_t^2 \delta^2 + \gamma_t^2)}\boldsymbol{U}_k^\perp \boldsymbol{U}_k^{\perp T}\boldsymbol{x}\right)}{} \\
&= -\frac{1}{\gamma_t^2}\left(\boldsymbol{x} - \frac{\sum_{k=1}^K \pi_k \mathcal{N}(\boldsymbol{x}; \boldsymbol{0}, s_t^2 \boldsymbol{U}_k \boldsymbol{U}_k^T + s_t^2 \delta^2 \boldsymbol{U}_k^\perp \boldsymbol{U}_k^{\perp T} + \gamma_t^2 \boldsymbol{I}_n)}{\sum_{k=1}^K \pi_k \mathcal{N}(\boldsymbol{x}; \boldsymbol{0}, s_t^2 \boldsymbol{U}_k \boldsymbol{U}_k^T + s_t^2 \delta^2 \boldsymbol{U}_k^\perp \boldsymbol{U}_k^{\perp T} + \gamma_t^2 \boldsymbol{I}_n)} \right. \\
&\qquad\qquad \left. \cdot \left(\frac{s_t^2}{s_t^2 + \gamma_t^2}\boldsymbol{U}_k \boldsymbol{U}_k^T \boldsymbol{x}\right) + \frac{s_t^2 \delta^2}{s_t^2 \delta^2 + \gamma_t^2}\boldsymbol{U}_k^\perp \boldsymbol{U}_k^{\perp T}\boldsymbol{x}\right).
\end{aligned}
$$

According to Tweedie's formula, we have

$$
\begin{aligned}
\mathbb{E}[\boldsymbol{x}_0|\boldsymbol{x}_t] &= \frac{\boldsymbol{x}_t + \gamma_t^2 \nabla \log p_t(\boldsymbol{x}_t)}{s_t} \\
&= \frac{s_t}{s_t^2 + \gamma_t^2}\frac{\sum_{k=1}^K \pi_k \mathcal{N}(\boldsymbol{x}; \boldsymbol{0}, s_t^2 \boldsymbol{U}_k \boldsymbol{U}_k^T + s_t^2 \delta^2 \boldsymbol{U}_k^\perp \boldsymbol{U}_k^{\perp T} + \gamma_t^2 \boldsymbol{I}_n)\boldsymbol{U}_k \boldsymbol{U}_k^T \boldsymbol{x}}{\mathcal{N}(\boldsymbol{x}; \boldsymbol{0}, s_t^2 \boldsymbol{U}_k \boldsymbol{U}_k^T + s_t^2 \delta^2 \boldsymbol{U}_k^\perp \boldsymbol{U}_k^{\perp T} + \gamma_t^2 \boldsymbol{I}_n)} \\
&\quad + \frac{s_t \delta^2}{s_t^2 \delta^2 + \gamma_t^2}\frac{\sum_{k=1}^K \pi_k \mathcal{N}(\boldsymbol{x}; \boldsymbol{0}, s_t^2 \boldsymbol{U}_k \boldsymbol{U}_k^T + s_t^2 \delta^2 \boldsymbol{U}_k^\perp \boldsymbol{U}_k^{\perp T} + \gamma_t^2 \boldsymbol{I}_n)\boldsymbol{U}_k^\perp \boldsymbol{U}_k^{\perp T}\boldsymbol{x}}{\mathcal{N}(\boldsymbol{x}; \boldsymbol{0}, s_t^2 \boldsymbol{U}_k \boldsymbol{U}_k^T + s_t^2 \delta^2 \boldsymbol{U}_k^\perp \boldsymbol{U}_k^{\perp T} + \gamma_t^2 \boldsymbol{I}_n)} \\
&= \frac{s_t}{s_t^2 + \gamma_t^2}\frac{\sum_{k=1}^K \pi_k \exp\left(\phi_t \|\boldsymbol{U}_k^T \boldsymbol{x}_t\|^2\right)\exp\left(\psi_t \|\boldsymbol{U}_k^{\perp T}\boldsymbol{x}_t\|^2\right)\boldsymbol{U}_k \boldsymbol{U}_k^T \boldsymbol{x}_t}{\sum_{k=1}^K \pi_k \exp\left(\phi_t \|\boldsymbol{U}_k^T \boldsymbol{x}_t\|^2\right)\exp\left(\psi_t \|\boldsymbol{U}_k^{\perp T}\boldsymbol{x}_t\|^2\right)} \\
&\quad + \frac{s_t \delta^2}{s_t^2 \delta^2 + \gamma_t^2}\frac{\sum_{k=1}^K \pi_k \exp\left(\phi_t \|\boldsymbol{U}_k^T \boldsymbol{x}_t\|^2\right)\exp\left(\psi_t \|\boldsymbol{U}_k^{\perp T}\boldsymbol{x}_t\|^2\right)\boldsymbol{U}_k^\perp \boldsymbol{U}_k^{\perp T}\boldsymbol{x}_t}{\sum_{k=1}^K \pi_k \exp\left(\phi_t \|\boldsymbol{U}_k^T \boldsymbol{x}_t\|^2\right)\exp\left(\psi_t \|\boldsymbol{U}_k^{\perp T}\boldsymbol{x}_t\|^2\right)},
\end{aligned}
$$

with $\phi_t = s_t^2/(2\gamma_t^2(s_t^2 + \gamma_t^2))$ and $\psi_t = s_t^2 \delta^2/(2\gamma_t^2(s_t^2 \delta^2 + \gamma_t^2))$. The final equality uses the pdf of multi-variant Gaussian and the matrix inversion lemma discussed earlier.

Now since $\pi_k$ is consistent for all $k$ and $s_t = 1$, we have

$$\mathbb{E}[\boldsymbol{x}_0|\boldsymbol{x}_t] = \sum_{k=1}^K w_k^\star(\boldsymbol{x}_t)\left(\frac{1}{1 + \sigma_t^2}\boldsymbol{U}_k \boldsymbol{U}_k^T + \frac{\delta^2}{\delta^2 + \sigma_t^2}\boldsymbol{U}_k^\perp \boldsymbol{U}_k^{\perp T}\right)\boldsymbol{x}_t$$

$$\text{where } w_k^\star(\boldsymbol{x}_t) := \frac{\exp\left(\frac{1}{2\sigma_t^2(1 + \sigma_t^2)}\|\boldsymbol{U}_k^T \boldsymbol{x}_t\|^2 + \frac{\delta^2}{2\sigma_t^2(\delta^2 + \sigma_t^2)}\|\boldsymbol{U}_k^{\perp T}\boldsymbol{x}_t\|^2\right)}{\sum_{k=1}^K \exp\left(\frac{1}{2\sigma_t^2(1 + \sigma_t^2)}\|\boldsymbol{U}_k^T \boldsymbol{x}_t\|^2 + \frac{\delta^2}{2\sigma_t^2(\delta^2 + \sigma_t^2)}\|\boldsymbol{U}_k^{\perp T}\boldsymbol{x}_t\|^2\right)}.$$

$\square$

### A.4.2 PROOF OF THEOREM 1

*Proof.* Following Equation (8) and Lemma 1, we can write

$$
\text{CSNR}(t, \hat{\boldsymbol{x}}^{\star}_{approx}) = \frac{\mathbb{E}_{\boldsymbol{x}_0}[\|\boldsymbol{U}_k\boldsymbol{U}_k^T\hat{\boldsymbol{x}}^{\star}_{approx}(\boldsymbol{x}_0,t)\|^2]}{\mathbb{E}_{\boldsymbol{x}_0}[\sum_{l\neq k}\|\boldsymbol{U}_l\boldsymbol{U}_l^T\hat{\boldsymbol{x}}^{\star}_{approx}(\boldsymbol{x}_0,t)\|^2]} = \frac{\mathbb{E}_{\boldsymbol{x}_0}[\|\boldsymbol{U}_k\boldsymbol{U}_k^T\hat{\boldsymbol{x}}^{\star}_{approx}(\boldsymbol{x}_0,t)\|^2]}{\sum_{l\neq k}\mathbb{E}_{\boldsymbol{x}_0}[\|\boldsymbol{U}_l\boldsymbol{U}_l^T\hat{\boldsymbol{x}}^{\star}_{approx}(\boldsymbol{x}_0,t)\|^2]}
$$

$$
= \frac{\left(\frac{\hat{w}_k}{1+\sigma_t^2} + \frac{(K-1)\delta^2\hat{w}_l}{\delta^2+\sigma_t^2}\right)^2 d}{(K-1)\left(\frac{\hat{w}_l}{1+\sigma_t^2} + \frac{\delta^2(\hat{w}_k+(K-2)\hat{w}_l)}{\delta^2+\sigma_t^2}\right)^2 \delta^2 d}
$$

$$
= \frac{1}{(K-1)\delta^2} \cdot \left(\frac{\hat{w}_k\delta^2 + \hat{w}_k\sigma_t^2 + (K-1)\delta^2\hat{w}_l + (K-1)\delta^2\hat{w}_l\sigma_t^2}{\hat{w}_l\delta^2 + \hat{w}_l\sigma_t^2 + \delta^2\hat{w}_k + (K-2)\delta^2\hat{w}_l + \delta^2\hat{w}_k\sigma_t^2 + (K-2)\delta^2\hat{w}_l\sigma_t^2}\right)^2
$$

$$
= \frac{1}{(K-1)\delta^2} \cdot \left(\frac{\delta^2 + \sigma_t^2\left(\hat{w}_k + (K-1)\delta^2\hat{w}_l\right)}{\delta^2 + \sigma_t^2\left(\hat{w}_l + \delta^2\hat{w}_k + (K-2)\delta^2\hat{w}_l\right)}\right)^2
$$

$$
= \frac{1}{(K-1)\delta^2} \cdot \left(\frac{1 + \frac{\sigma_t^2}{\delta^2}\left((1-\delta^2)\hat{w}_k + \delta^2(\hat{w}_k + (K-1)\hat{w}_l)\right)}{1 + \frac{\sigma_t^2}{\delta^2}\left((1-\delta^2)\hat{w}_l + \delta^2(\hat{w}_l + \hat{w}_k + (K-2)\hat{w}_l)\right)}\right)^2
$$

$$
= \frac{1}{(K-1)\delta^2} \cdot \left(\frac{1 + \frac{\sigma_t^2}{\delta^2}\left((1-\delta^2)\hat{w}_k + \delta^2\right)}{1 + \frac{\sigma_t^2}{\delta^2}\left((1-\delta^2)\hat{w}_l + \delta^2\right)}\right)^2
$$

$$
= \frac{1}{(K-1)\delta^2} \cdot \left(\frac{1 + \frac{\sigma_t^2}{\delta^2}h(\hat{w}_k,\delta)}{1 + \frac{\sigma_t^2}{\delta^2}h(\hat{w}_l,\delta)}\right)^2
$$

where $h(w,\delta) := (1-\delta^2)w + \delta^2$. $\qquad\square$

**Lemma 1.** *With the set up of a K-class* MoLRG *data distribution as defined in (4), consider the following the function:*

$$
\hat{\boldsymbol{x}}^{\star}_{approx}(\boldsymbol{x},t) = \sum_{k=1}^{K}\hat{w}_k(\boldsymbol{x})\left(\frac{1}{1+\sigma_t^2}\boldsymbol{U}_k\boldsymbol{U}_k^T + \frac{\delta^2}{\delta^2+\sigma_t^2}\boldsymbol{U}_k^{\perp}\boldsymbol{U}_k^{\perp T}\right)\boldsymbol{x}, \tag{11}
$$

$$
\text{where } \hat{w}_k(\boldsymbol{x}) := \frac{\exp\left(\mathbb{E}_{\boldsymbol{x}}[g_k(\boldsymbol{x},t)]\right)}{\sum_{k=1}^{K}\exp\left(\mathbb{E}_{\boldsymbol{x}}[g_k(\boldsymbol{x},t)]\right)}, \tag{12}
$$

$$
\text{and } g_k(\boldsymbol{x}) = \frac{1}{2\sigma_t^2(1+\sigma_t^2)}\|\boldsymbol{U}_k^T\boldsymbol{x}\|^2 + \frac{\delta^2}{2\sigma_t^2(\delta^2+\sigma_t^2)}\|\boldsymbol{U}_k^{\perp T}\boldsymbol{x}\|^2. \tag{13}
$$

*I.e., we consider a simplified version of the expected posterior mean as in equation 5 by taking expectation of $g_k(\boldsymbol{x})$ prior to the softmax operation. Under this setting, for any clean $\boldsymbol{x}_0$ from class $k$ (i.e., $\boldsymbol{x}_0 = \boldsymbol{U}_k\boldsymbol{a}_i + b\boldsymbol{U}_k^{\perp}\boldsymbol{e}_i$), we have:*

$$
\mathbb{E}_{\boldsymbol{x}_0}[\|\boldsymbol{U}_k\boldsymbol{U}_k^T\hat{\boldsymbol{x}}^{\star}_{approx}(\boldsymbol{x}_0,t)\|^2] = \left(\frac{\hat{w}_k}{1+\sigma_t^2} + \frac{(K-1)\delta^2\hat{w}_l}{\delta^2+\sigma_t^2}\right)^2 d \tag{14}
$$

$$
\mathbb{E}_{\boldsymbol{x}_0}[\|\boldsymbol{U}_l\boldsymbol{U}_l^T\hat{\boldsymbol{x}}^{\star}_{approx}(\boldsymbol{x}_0,t)\|^2] = \left(\frac{\hat{w}_l}{1+\sigma_t^2} + \frac{\delta^2(\hat{w}_k+(K-2)\hat{w}_l)}{\delta^2+\sigma_t^2}\right)^2 \delta^2 d \tag{15}
$$

$$
\mathbb{E}_{\boldsymbol{x}_0}[\|\boldsymbol{U}_{\perp}\boldsymbol{U}_{\perp}^T\hat{\boldsymbol{x}}^{\star}_{approx}(\boldsymbol{x}_0,t)\|^2] = \frac{\delta^6(n-kd)}{(\delta^2+\sigma_t^2)^2} \tag{16}
$$

$$\mathbb{E}[\|\hat{\boldsymbol{x}}_{approx}^{\star}(\boldsymbol{x}_0, t)\|^2] = \underbrace{\left(\frac{\hat{w}_k}{1 + \sigma_t^2} + \frac{(K-1)\delta^2 \hat{w}_l}{\delta^2 + \sigma_t^2}\right)^2 d}_{\mathbb{E}\|\boldsymbol{U}_k \boldsymbol{U}_k^T \hat{\boldsymbol{x}}_{approx}^{\star}(\boldsymbol{x}_0, t)\|^2]}$$

$$+ \underbrace{(K-1)\left(\frac{\hat{w}_l}{1 + \sigma_t^2} + \frac{\delta^2(\hat{w}_k + (K-2)\hat{w}_l)}{\delta^2 + \sigma_t^2}\right)^2 \delta^2 d}_{\mathbb{E}[\sum_{l \neq k}^{K} \boldsymbol{U}_l \boldsymbol{U}_l^T \hat{\boldsymbol{x}}_{approx}^{\star}(\boldsymbol{x}_0, t)\|^2]} + \underbrace{\frac{\delta^6(n - Kd)}{(\delta^2 + \sigma_t^2)^2}}_{\mathbb{E}[\|\boldsymbol{U}_{\perp} \boldsymbol{U}_{\perp}^T \hat{\boldsymbol{x}}_{approx}^{\star}(\boldsymbol{x}_0, t)\|^2]}$$

$$\tag{17}$$

*and*

$$\hat{w}_k := \hat{w}_k(\boldsymbol{x}_0) = \frac{\exp\left(\frac{d}{2\sigma_t^2(1+\sigma_t^2)} + \frac{\delta^4 D}{2\sigma_t^2(\delta^2+\sigma_t^2)}\right)}{\exp\left(\frac{d}{2\sigma_t^2(1+\sigma_t^2)} + \frac{\delta^4 D}{2\sigma_t^2(\delta^2+\sigma_t^2)}\right) + (K-1)\exp\left(\frac{\delta^2 d}{2\sigma_t^2(1+\sigma_t^2)} + \frac{\delta^2 d + \delta^4(D-d)}{2\sigma_t^2(\delta^2+\sigma_t^2)}\right)},$$

$$\hat{w}_l := \hat{w}_l(\boldsymbol{x}_0) = \frac{\exp\left(\frac{\delta^2 d}{2\sigma_t^2(1+\sigma_t^2)} + \frac{\delta^2 d + \delta^4(D-d)}{2\sigma_t^2(\delta^2+\sigma_t^2)}\right)}{\exp\left(\frac{d}{2\sigma_t^2(1+\sigma_t^2)} + \frac{\delta^4 D}{2\sigma_t^2(\delta^2+\sigma_t^2)}\right) + (K-1)\exp\left(\frac{\delta^2 d}{2\sigma_t^2(1+\sigma_t^2)} + \frac{\delta^2 d + \delta^4(D-d)}{2\sigma_t^2(\delta^2+\sigma_t^2)}\right)}$$

$$\tag{18}$$

*for all class index $l \neq k$.*

*Proof.* Throughout the proof, we use the following notation for slices of vectors.

$$\boldsymbol{e}_i[a : b] \qquad \text{Slices of vector } \boldsymbol{e}_i \text{ from } a\text{th entry to } b\text{th entry.}$$

We begin with the softmax terms. Since each class has its unique disjoint subspace, it suffices to consider $g_k(\boldsymbol{x}_0, t)$ and $g_l(\boldsymbol{x}_0, t)$ for any $l \neq k$. Let $a_t = \frac{1}{2\sigma_t^2(1+\sigma_t^2)}$ and $c_t = \frac{\delta^2}{2\sigma_t^2(\delta^2+\sigma_t^2)}$, we have:

$$\begin{aligned}
\mathbb{E}[g_k(\boldsymbol{x}_0, t)] &= \mathbb{E}[a_t \|\boldsymbol{U}_k^T \boldsymbol{x}_0\|^2 + c_t \|\boldsymbol{U}_k^{\perp T} \boldsymbol{x}_0\|^2] \\
&= \mathbb{E}[a_t \|\boldsymbol{U}_k^T(\boldsymbol{U}_k \boldsymbol{a}_i + b\boldsymbol{U}_k^{\perp} \boldsymbol{e}_i)\|^2] + \mathbb{E}[c_t \|\boldsymbol{U}_k^{\perp T}(\boldsymbol{U}_k \boldsymbol{a}_i + b\boldsymbol{U}_k^{\perp} \boldsymbol{e}_i)\|^2] \\
&= \mathbb{E}[a_t \|\boldsymbol{a}_i\|^2] + \mathbb{E}[c_t \|b\boldsymbol{e}_i\|^2] \\
&= a_t d + c_t \delta^2 D
\end{aligned}$$

where the last equality follows from $\boldsymbol{a}_i \overset{i.i.d.}{\sim} \mathcal{N}(\boldsymbol{0}, \boldsymbol{I}_d)$ and $\boldsymbol{e}_i \overset{i.i.d.}{\sim} \mathcal{N}(\boldsymbol{0}, \boldsymbol{I}_D)$.

Without loss of generality, assume the $j = k + 1$, we have:

$$\begin{aligned}
\mathbb{E}[g_l(\boldsymbol{x}_0, t)] &= \mathbb{E}[a_t \|\boldsymbol{U}_l^T \boldsymbol{x}_0\|^2 + c_t \|\boldsymbol{U}_l^{\perp T} \boldsymbol{x}_0\|^2] \\
&= \mathbb{E}[a_t \|\boldsymbol{U}_l^T(\boldsymbol{U}_k \boldsymbol{a}_i + b\boldsymbol{U}_k^{\perp} \boldsymbol{e}_i)\|^2] + \mathbb{E}[c_t \|\boldsymbol{U}_l^{\perp T}(\boldsymbol{U}_k \boldsymbol{a}_i + b\boldsymbol{U}_k^{\perp} \boldsymbol{e}_i)\|^2] \\
&= \mathbb{E}[a_t \|b\boldsymbol{e}_i[1 : d]\|^2] + \mathbb{E}\left[c_t \left\|\begin{bmatrix} \boldsymbol{a}_i \\ \boldsymbol{0} \in \mathbb{R}^{D-d} \end{bmatrix} + b\begin{bmatrix} \boldsymbol{0} \in \mathbb{R}^d \\ \boldsymbol{e}_i[d : D]] \end{bmatrix}\right\|^2\right] \\
&= a_t \delta^2 d + c_t(d + \delta^2(D - d))
\end{aligned}$$

Plug $a_t$ and $b_t$ back with the exponentials, we get $\hat{w}_k$ and $\hat{w}_l$.

Now we prove (14):

$$\boldsymbol{U}_k\boldsymbol{U}_k^T\hat{\boldsymbol{x}}_{approx}^\star(\boldsymbol{x}_0,t) = \hat{w}_k\boldsymbol{U}_k\boldsymbol{U}_k^T\left(\frac{1}{1+\sigma_t^2}\boldsymbol{U}_k\boldsymbol{U}_k^T + \frac{\delta^2}{\delta^2+\sigma_t^2}\boldsymbol{U}_k^\perp\boldsymbol{U}_k^{\perp T}\right)\boldsymbol{x}_0$$

$$+ \sum_{l\neq k}\hat{w}_l\boldsymbol{U}_k\boldsymbol{U}_k^T\left(\frac{1}{1+\sigma_t^2}\boldsymbol{U}_l\boldsymbol{U}_l^T + \frac{\delta^2}{\delta^2+\sigma_t^2}\boldsymbol{U}_l^\perp\boldsymbol{U}_l^{\perp T}\right)\boldsymbol{x}_0$$

$$= \hat{w}_k\left(\frac{1}{1+\sigma_t^2}\boldsymbol{U}_k\boldsymbol{U}_k^T\boldsymbol{x}_0\right) + \sum_{l\neq k}\hat{w}_l\left(\frac{\delta^2}{\delta^2+\sigma_t^2}\boldsymbol{U}_k\boldsymbol{U}_k^T\boldsymbol{x}_0\right)$$

$$= \left(\frac{\hat{w}_k}{1+\sigma_t^2} + \frac{(K-1)\,\delta^2\hat{w}_l}{\delta^2+\sigma_t^2}\right)\boldsymbol{U}_k\boldsymbol{U}_k^T(\boldsymbol{U}_k\boldsymbol{a}_i + b\boldsymbol{U}_k^\perp\boldsymbol{e}_i)$$

$$= \left(\frac{\hat{w}_k}{1+\sigma_t^2} + \frac{(K-1)\,\delta^2\hat{w}_l}{\delta^2+\sigma_t^2}\right)\boldsymbol{U}_k\boldsymbol{a}_i$$

Since $\boldsymbol{U}_k \in \mathcal{O}^{n\times d}$:

$$\mathbb{E}[\|\boldsymbol{U}_k\boldsymbol{U}_k^T\hat{\boldsymbol{x}}_{approx}^\star(\boldsymbol{x}_0,t)\|^2] = \left(\frac{\hat{w}_k}{1+\sigma_t^2} + \frac{(K-1)\,\delta^2\hat{w}_l}{\delta^2+\sigma_t^2}\right)^2 d$$

and similarly for (15):

$$\boldsymbol{U}_l\boldsymbol{U}_l^T\hat{\boldsymbol{x}}_{approx}^\star(\boldsymbol{x}_0,t) = \hat{w}_k\boldsymbol{U}_l\boldsymbol{U}_l^T\left(\frac{1}{1+\sigma_t^2}\boldsymbol{U}_k\boldsymbol{U}_k^T + \frac{\delta^2}{\delta^2+\sigma_t^2}\boldsymbol{U}_k^\perp\boldsymbol{U}_k^{\perp T}\right)\boldsymbol{x}_0$$

$$+ \hat{w}_l\boldsymbol{U}_l\boldsymbol{U}_l^T\left(\frac{1}{1+\sigma_t^2}\boldsymbol{U}_l\boldsymbol{U}_l^T + \frac{\delta^2}{\delta^2+\sigma_t^2}\boldsymbol{U}_l^\perp\boldsymbol{U}_l^{\perp T}\right)\boldsymbol{x}_0$$

$$+ \sum_{j\neq k,l}\hat{w}_j\boldsymbol{U}_l\boldsymbol{U}_l^T\left(\frac{1}{1+\sigma_t^2}\boldsymbol{U}_j\boldsymbol{U}_j^T + \frac{\delta^2}{\delta^2+\sigma_t^2}\boldsymbol{U}_j^\perp\boldsymbol{U}_j^{\perp T}\right)\boldsymbol{x}_0$$

$$= \hat{w}_k\left(\frac{\delta^2}{\delta^2+\sigma_t^2}\boldsymbol{U}_l\boldsymbol{U}_l^T\boldsymbol{x}_0\right) + \hat{w}_l\left(\frac{1}{1+\sigma_t^2}\boldsymbol{U}_l\boldsymbol{U}_l^T\boldsymbol{x}_0\right) + \sum_{j\neq k,l}\hat{w}_j\left(\frac{\delta^2}{\delta^2+\sigma_t^2}\boldsymbol{U}_l\boldsymbol{U}_l^T\boldsymbol{x}_0\right)$$

$$= \left(\frac{\hat{w}_l}{1+\sigma_t^2} + \frac{\delta^2(\hat{w}_k + (K-2)\hat{w}_j)}{\delta^2+\sigma_t^2}\right)\boldsymbol{U}_l\boldsymbol{U}_l^T(\boldsymbol{U}_k\boldsymbol{a}_i + b\boldsymbol{U}_k^\perp\boldsymbol{e}_i)$$

$$= \left(\frac{\hat{w}_l}{1+\sigma_t^2} + \frac{\delta^2(\hat{w}_k + (K-2)\hat{w}_l)}{\delta^2+\sigma_t^2}\right)b\boldsymbol{U}_l\boldsymbol{e}_i[1:d]$$

where the third equality follows since $\hat{w}_j = \hat{w}_l$ for all $j \neq k, l$. Further, we have:

$$\mathbb{E}[\|\boldsymbol{U}_l\boldsymbol{U}_l^T\hat{\boldsymbol{x}}_{approx}^\star(\boldsymbol{x}_0,t)\|^2] = \left(\frac{\hat{w}_l}{1+\sigma_t^2} + \frac{\delta^2(\hat{w}_k + (K-2)\hat{w}_l)}{\delta^2+\sigma_t^2}\right)^2\delta^2 d$$

Next, we consider (16):

$$\boldsymbol{U}_\perp\boldsymbol{U}_\perp^T\hat{\boldsymbol{x}}_{approx}^\star(\boldsymbol{x}_0,t) = \hat{w}_k\boldsymbol{U}_\perp\boldsymbol{U}_\perp^T\left(\frac{1}{1+\sigma_t^2}\boldsymbol{U}_k\boldsymbol{U}_k^T + \frac{\delta^2}{\delta^2+\sigma_t^2}\boldsymbol{U}_k^\perp\boldsymbol{U}_k^{\perp T}\right)\boldsymbol{x}_0$$

$$+ \sum_{l\neq k}\hat{w}_l\boldsymbol{U}_\perp\boldsymbol{U}_\perp^T\left(\frac{1}{1+\sigma_t^2}\boldsymbol{U}_l\boldsymbol{U}_l^T + \frac{\delta^2}{\delta^2+\sigma_t^2}\boldsymbol{U}_l^\perp\boldsymbol{U}_l^{\perp T}\right)\boldsymbol{x}_0$$

$$= \hat{w}_k\left(\frac{\delta^2}{\delta^2+\sigma_t^2}\boldsymbol{U}_\perp\boldsymbol{U}_\perp^T\boldsymbol{x}_0\right) + \sum_{l\neq k}\hat{w}_l\left(\frac{\delta^2}{\delta^2+\sigma_t^2}\boldsymbol{U}_\perp\boldsymbol{U}_\perp^T\boldsymbol{x}_0\right)$$

$$= \frac{\delta^2}{\delta^2+\sigma_t^2}\boldsymbol{U}_\perp\boldsymbol{U}_\perp^T(\boldsymbol{U}_k\boldsymbol{a}_i + b\boldsymbol{U}_k^\perp\boldsymbol{e}_i)$$

$$= \frac{\delta^3}{\delta^2+\sigma_t^2}\boldsymbol{U}_\perp\boldsymbol{e}_i[(K-1)d:D]$$

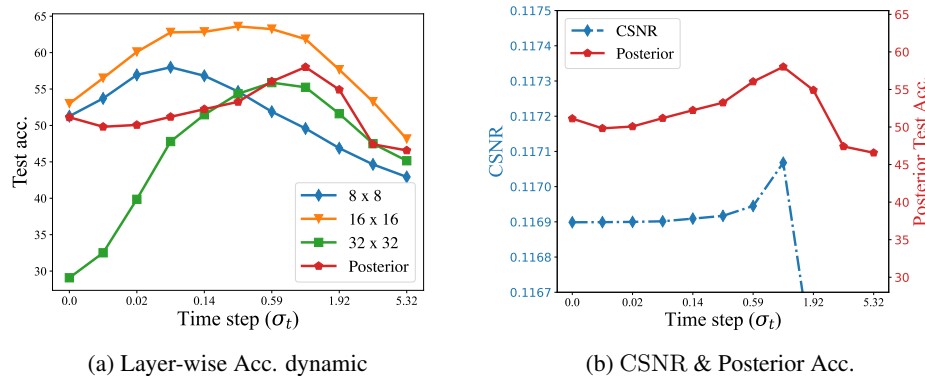

(a) Layer-wise Acc. dynamic        (b) CSNR & Posterior Acc.

Figure 10: **Investigation of the layer-wise dynamic of diffusion-based representation learning.** We use DDPM pre-trained diffusion model on CIFAR10 and plot the test accuracy achieved by its features at various resolutions in (a) and the posterior probing accuracy and CSNR in (b).

Hence:

$$\mathbb{E}[\|\boldsymbol{U}_\perp \boldsymbol{U}_\perp^T \hat{\boldsymbol{x}}_{approx}^\star(\boldsymbol{x}_0, t)\|^2] = \frac{\delta^6(n - Kd)}{(\delta^2 + \sigma_t^2)^2}$$

Lastly, we prove (17). Given that the subspaces of all classes and the complement space are both orthonormal and mutually orthogonal, we can write:

$$\mathbb{E}[\|\hat{\boldsymbol{x}}_{approx}^\star(\boldsymbol{x}_0, t)\|^2] = \mathbb{E}[\|\boldsymbol{U}_k \boldsymbol{U}_k^T \hat{\boldsymbol{x}}_{approx}^\star(\boldsymbol{x}_0, t)\|^2] + \mathbb{E}[\sum_{l \neq k} \|\boldsymbol{U}_l \boldsymbol{U}_l^T \hat{\boldsymbol{x}}_{approx}^\star(\boldsymbol{x}_0, t)\|^2] + \mathbb{E}[\|\boldsymbol{U}_\perp \boldsymbol{U}_\perp^T \hat{\boldsymbol{x}}_{approx}^\star(\boldsymbol{x}_0, t)\|^2]$$

Combine terms, we get:

$$\mathbb{E}[\|\hat{\boldsymbol{x}}_{approx}^\star(\boldsymbol{x}_0, t)\|^2] = \left( \frac{\hat{w}_k}{1 + \sigma_t^2} + \frac{(K-1)\delta^2 \hat{w}_l}{\delta^2 + \sigma_t^2} \right)^2 d$$
$$+ (K-1) \left( \frac{\hat{w}_l}{1 + \sigma_t^2} + \frac{\delta^2(\hat{w}_k + (K-2)\hat{w}_l)}{\delta^2 + \sigma_t^2} \right)^2 \delta^2 d + \frac{\delta^6(n - Kd)}{(\delta^2 + \sigma_t^2)^2}.$$

$\square$

## A.5 NEWLY ADDED EXPERIMENTS

**Investigation of Layer-Wise Dynamics in Diffusion-Based Representation Learning.** In the main body of the paper, we focus on the features extracted from the UNet bottleneck layer. In this subsection, we extend our analysis to investigate the layer-wise representation dynamics within a diffusion model. Using the pre-trained DDPM on CIFAR10 from Figure 10, we extract features from the UNet decoder at various resolutions. Since each resolution contains multiple blocks, we consistently select the first block with residual connections from the UNet encoder at each resolution. The test accuracy results of these features are shown in Figure 10(a), where we observe a progressive shift in the accuracy peak from shallow to deeper layers, eventually aligning with the posterior test accuracy. Additionally, we plot the CSNR in Figure 10(b) and find that its trend also demonstrates an unimodal curve.

**Posterior Estimation Quality: Comparison Between $x_t$ and $x_0$ as inputs.** In Figure 1(c)-(d) and Figure 8, we have demonstrated that using clean images $x_0$ as inputs to the diffusion model achieves on-par or superior representation learning performance compared to using noisy images $x_t$, particularly under high noise regimes. In this subsection, we show that this improved representation learning performance directly reflects the superior posterior estimation quality of clean inputs. In Figure 11, using the pre-trained DDPM on CIFAR10 from Figure 10, we visualizes the posterior

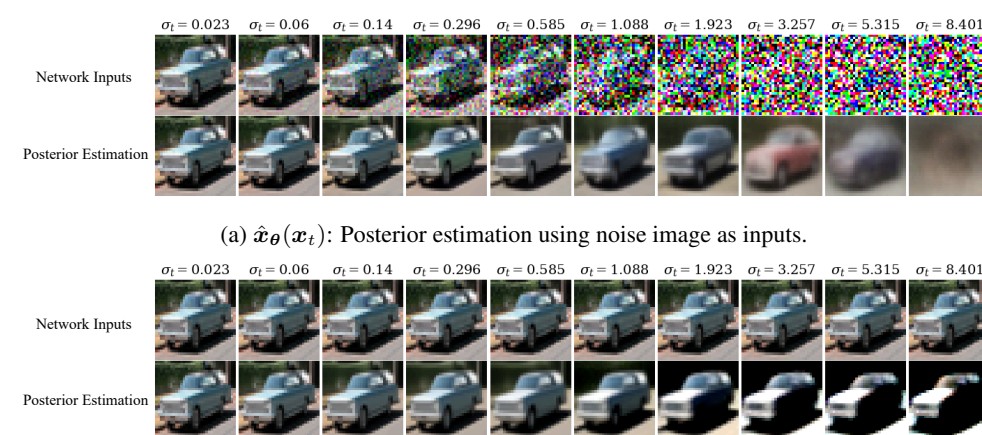

(a) $\hat{x}_{\theta}(x_t)$: Posterior estimation using noise image as inputs.

(b) $\hat{x}_{\theta}(x_0)$: Posterior estimation using clean image as inputs.

Figure 11: **Comparison of posterior estimation using clean and noisy inputs.** We employ a DDPM pre-trained diffusion model on CIFAR10 to visualize the posterior estimation for both clean inputs and noisy inputs as a function of the noise scale $\sigma_t$.

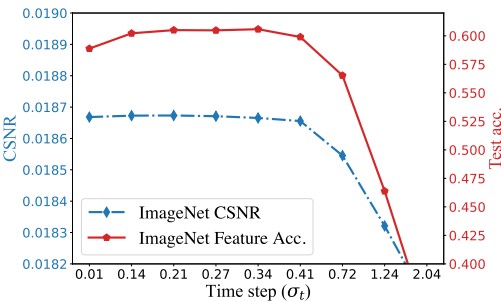

Figure 12: **Additional results on ImageNet.** We employ a pre-trained DiT diffusion model on ImageNet. Intermediate features used for linear probing are extracted from the 1/2-L layer. The CSNR metric is computed on the latent codes of clean images, obtained after processing them through a VAE.

estimation results for clean inputs ($\hat{x}_{\theta}(x_0)$) and noisy inputs ($\hat{x}_{\theta}(x_t)$) across varying noise scales $\sigma_t$. As shown, the posterior estimation for clean and noisy inputs appears similar when the noise scale is small. However, as the noise scale increases, the posterior estimation from clean images exhibits superior visual quality, aligning with the better feature probing accuracy reported in the main body of the paper.

**Additional experiment on ImageNet Deng et al. (2009).** We use a DiT-XL (Peebles & Xie, 2023) pre-trained on ImageNet and follow the settings in previous work (Chen et al., 2024b). Specifically, we extract features from the 1/2-L layer, treated as the bottleneck layer, and apply mean-pooling over all tokens for linear probing. Due to computational constraints, we limit our analysis to the 100 class labels used in miniImageNet (Vinyals et al., 2016). For computing CSNR, since DiT employs latent diffusion, we compute CSNR on the latent space. This involves first passing the images through a VAE to obtain the latent representation, flattening the output, and then computing the basis from the SVD of the flattened latent vectors. As shown in Figure 12, both the feature probing accuracy and CSNR exhibit a similar curve, consistent with findings on other datasets and network architectures discussed in the main body of the paper.

**Additional experiments on `MoLRG` data.** In Figure 4, we can observe that CSNR of both the learned DAE and diffusion model have a gap to the optimal CSNR. We hypothesize that the

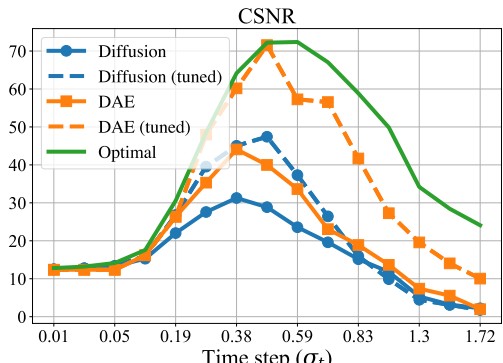

Figure 13: CSNR **comparison on `MoLRG` data.** Using the same model and data as in Figure 4, we plot the CSNR results for the (tuned) DAEs and (tuned) diffusion model. Solid lines represent normal training results (as in Figure 4), while dashed lines indicate results with more nuanced optimization strategy for improved performance.

discrepancy between the trained network and the optimal solution may arise from the following two factors:

- **Network Capacity.** A single DAE is tailored to handle a specific noise scale, enabling its CSNR to closely align with the optimal CSNR across multiple noise scales. Conversely, the diffusion model must simultaneously accommodate all noise scales, which compromises its performance on individual noise scales. To test this hypothesis, we conducted an experiment in which we tuned the learning rate and extended the training duration to 1000 epochs. The results, shown in Figure 13, reveal that while the tuned diffusion model outperforms its untuned counterpart, it still exhibits a substantial gap compared to the optimal CSNR, thus verifies the conjecture.

- **Optimization Difficulty.** As described in Equation equation 5, the optimal posterior function requires projecting $x_t$ onto different subspaces. At higher noise levels, the magnitude of this projection diminishes (since $\sigma_t$ appears only in the denominator), making optimization increasingly challenging. To explore this hypothesis, we employed more nuanced optimization strategies for the DAE models. These include increasing training epochs (from 200 to 1000), decreasing learning rates (from $1e^{-3}$ to $1e^{-4}$), and scaling down the initialization magnitude as the noise level increases. While these strategies effectively drive the CSNR closer to the optimal CSNR for small noise scales, a persistent gap remains at larger noise scales due to the enlarged optimization difficulty.

Furthermore, we conduct a preliminary study to investigate the potential of CSNR as a metric for model comparison by plotting the posterior probing accuracy and intermediate probing accuracy for the (tuned) DAEs and (tuned) diffusion models in Figure 14. As shown in the plot, CSNR is directly linked to posterior accuracy, with higher CSNR values correlating with improved posterior accuracy. Regarding to the feature probing accuracy, although comparing DAEs with diffusion models in this case is challenging due to the weight-sharing mechanism discussed in Section 4.1, we can still observe CSNR serves as a reliable metric for reflecting feature probing accuracy within the same model (e.g., tuned DAEs and diffusion models compared to their untuned counterparts)).

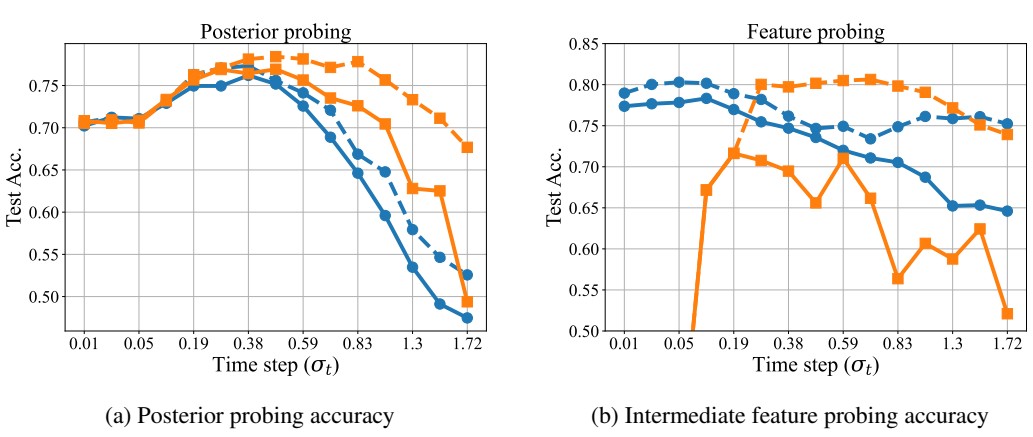

(a) Posterior probing accuracy     (b) Intermediate feature probing accuracy

Figure 14: **Connection between** CSNR **and posterior/intermediate feature probing accuracy.** We use the DAE and diffusion models trained for Figure 13 and plot the corresponding posterior and intermediate feature probing accuracy.

