# OpenReview forum: "Understanding Diffusion-based Representation Learning via Low-Dimensional Modeling"
_ICLR.cc/2025/Conference — Submitted to ICLR 2025_

### Official Review · Reviewer_ANHe · 2024-10-21

**Soundness:** 3
**Presentation:** 2
**Contribution:** 1
**Rating:** 3
**Confidence:** 4

**Summary:**

This work evaluates the potential of representation learning with diffusion models. The authors evaluate the connection between the quality of posterior estimation and the quality of learned representations, extending this analysis to different scenarios.

**Strengths:**

- In Section 4.1, the authors present an interesting and insightful analysis of the effect of diffusion training with multiple levels of noise passed through the same model on the learned representations. It is intriguing to see that diffusion models have relatively steady representations across different denoising timesteps.

**Weaknesses:**

- Several results presented in this work are not novel. For example, the evaluation presented in Figure 1 was already discussed in [1] and [2]. As noted by authors, the fact that representation learning dynamic captures a “fine-to-coarse” shift with the increased amount of noise was already noted in DAE works [3] and [4]
- I fail to see the significance of the contribution “Linking posterior estimation ability of diffusion models to representation learning”. Isn’t this observation a straightforward implication of the fact that samples at the late diffusion steps are, by definition, more noisy, which results in lower quality of the posterior estimation and higher entropy when analyzing predictions from the linear probe?
- The “representation learning” capability evaluation is very limited to the simple linear probing task. This significantly limits the significance of the presented results. Extending the analysis with other SSL tasks would strengthen the submission.
- Editorial: I’m lost in the presentation of this work as in Sec. 2.3 it is written that “since diffusion models tend to memorize the training data instead of learning underlying data distribution when the training dataset is small (Zhang et al., 2023), we focus on the case where sufficient training data is available throughout our analysis in Section 3”, yet there Figure 2 on the second page seems to be a reproduction of work by Zhang et al. The analysis related to Figure 2 is presented on the last page


[1] Xiang, Weilai, et al. "Denoising diffusion autoencoders are unified self-supervised learners." Proceedings of the IEEE/CVF International Conference on Computer Vision. 2023.
[2] Deja, Kamil, Tomasz Trzciński, and Jakub M. Tomczak. "Learning data representations with joint diffusion models." Joint European Conference on Machine Learning and Knowledge Discovery in Databases. Cham: Springer Nature Switzerland, 2023.
[3] Choi, Jooyoung, et al. "Perception prioritized training of diffusion models." Proceedings of the IEEE/CVF Conference on Computer Vision and Pattern Recognition. 2022.
[4] Wang, Binxu, and John J. Vastola. "Diffusion models generate images like painters: an analytical theory of outline first, details later." arXiv preprint arXiv:2303.02490 (2023).

**Questions:**

- As indicated in line 192, in a setup with clean images used being used for classification, the diffusion model is imputed with clear image and timestep t, “where t serves solely as an indicator of the noise level for diffusion model to adopt during feature extraction.” Isn’t this setup out of distribution for a model, as the diffusion model is never trained with a clear image or conditioned with a timestep equal to zero?
- Why limiting the evaluated features to only those encoded in the bottleneck layer of the UNet architecture? A lot of information relevant to classification (such as high-frequency features) might be passed through the residual connections.

---

> ### Author Response · Authors · 2024-11-22
> **Author Response [1/2]**
>
> We thank the reviewer for the careful review. In the following, we address the reviewer’s comments one-by-one.
>
> > Several results presented in this work are not novel. For example, the evaluation presented in Figure 1 was already discussed in [1] and [2]. As noted by authors, the fact that representation learning dynamic captures a “fine-to-coarse” shift with the increased amount of noise was already noted in DAE works [3] and [4]
>
> **Answer**: We thank the reviewer for their question and appreciate the opportunity to clarify. It seems there may be a misunderstanding regarding the focus of our paper. Figure 1 mainly serves to motivate our study of the unimodal representation curve, and we have never claimed it as a contribution of this work. Furthermore, we have appropriately cited and acknowledged the results referenced by the reviewer in the introduction.
>
> As highlighted at the bottom of Page 2, our work establishes the first theoretical framework to characterize the dynamics of representation learning. Building on the assumption that the data follows a mixture of low-rank Gaussians, we introduce metrics for measuring the representation quality based on the posterior mean estimation. We provided an explanation for why the optimal representation quality is achieved at an intermediate noise level, not at the clean image as the “fine-to-coarse” shift may suggest.
>
> > I fail to see the significance of the contribution “Linking posterior estimation ability of diffusion models to representation learning”. Isn’t this observation a straightforward implication of the fact that samples at the late diffusion steps are, by definition, more noisy, which results in lower quality of the posterior estimation and higher entropy when analyzing predictions from the linear probe?
>
> **Answer**: We appreciate the reviewer’s question and would like to clarify potential misinterpretations.
>
> As shown in Figure 1, both the posterior accuracy and feature accuracy follow unimodal curves as a function of noise. While the monotonic behavior in the high-noise regime aligns with the reviewer’s intuition, the unimodal pattern in the low-noise regime cannot be explained by this argument, making it a highly nontrivial phenomenon.
>
> Our work provides the first theoretical explanation for this unimodal behavior by connecting posterior estimation to representation learning. Unlike prior empirical studies on diffusion-based representation learning, which lacked theoretical grounding, we establish a foundational framework that explains these dynamics. This contribution, particularly the characterization of representation learning's unimodal dynamic behavior across different noise levels, highlights the significance of our study.
>
> > The “representation learning” capability evaluation is very limited to the simple linear probing task. This significantly limits the significance of the presented results. Extending the analysis with other SSL tasks would strengthen the submission.
>
> **Answer**: We thank the reviewer for the valuable suggestion. We would like to note that:
>
> * The unimodal representation dynamic has been observed across various representation learning tasks beyond classification, such as segmentation [1] and image correspondence [2].
>
> * In our work, given that our primary focus is not empirical, and due to current limitations in computational resources on our end, we have chosen classification tasks as the main proxy for studying representation learning, following many prior works (e.g., [3][4]).
>
> * In our updated manuscript, we have included an ImageNet experiment using a pre-trained DiT model, which investigates larger datasets and network architectures. Details of this experiment can be found in Appendix A.5 (Figure 12).
>
> That said, we agree with the reviewer that experiments on other tasks would provide additional valuable insights, and we plan to explore these directions further after the rebuttal period.
>
> [1] Baranchuk et al; Label-Efficient Semantic Segmentation with Diffusion Models.
> [2] Tang et al; Emergent Correspondence from Image Diffusion.
> [3] Xiang et al; Denoising Diffusion Autoencoders are Unified Self-supervised Learners.
> [4] Chen et al; Deconstructing Denoising Diffusion Models for Self-Supervised Learning.

---

> ### Author Response · Authors · 2024-11-22
> **Author Response [2/2]**
>
> > Editorial: I’m lost in the presentation of this work as in Sec. 2.3 it is written that “since diffusion models tend to memorize the training data instead of learning underlying data distribution when the training dataset is small (Zhang et al., 2023), we focus on the case where sufficient training data is available throughout our analysis in Section 3”, yet there Figure 2 on the second page seems to be a reproduction of work by Zhang et al. The analysis related to Figure 2 is presented on the last page
>
> **Answer**: We thank the reviewer for their question and would like to clarify our findings. Figure 2 highlights the relationship between representation learning and distribution approximation, a connection not addressed in Zhang et al. (2023).
> Our results in Figures 2(a) and 2(b) show that diffusion models learn effective representations only when they approximate the underlying data distribution. Specifically, Figure 2(b) demonstrates a transition in linear probing accuracy, from minimal performance to high accuracy, as the number of training samples increases. This transition aligns with the transition from memorization to generalization observed in Figure 2(a) and Zhang et al. (2023). The correspondence between these trends highlights that: (i) meaningful representations emerge in the generalization regime where diffusion models learn the underlying distributions, and (ii) no meaningful representations are learned in the memorization regime.
>
> Our study emphasizes the critical role of distribution learning in enabling effective representation learning, which can be achieved with sufficient training data.
>
> > As indicated in line 192, in a setup with clean images used being used for classification, the diffusion model is imputed with clear image and timestep t, “where t serves solely as an indicator of the noise level for diffusion model to adopt during feature extraction.” Isn’t this setup out of distribution for a model, as the diffusion model is never trained with a clear image or conditioned with a timestep equal to zero?
>
> **Answer**: We appreciate the reviewer for the question and we refer the reviewer to our topmost response to all reviewers for clarification on this point.
>
> > Why limiting the evaluated features to only those encoded in the bottleneck layer of the UNet architecture? A lot of information relevant to classification (such as high-frequency features) might be passed through the residual connections.
>
> **Answer**: We thank the reviewer for the valuable suggestion. In our study, we focused on the bottleneck layer to evaluate representation quality across timesteps rather than across layers, as the bottleneck layer is both commonly used and straightforward to analyze for this purpose.
>
> In response to the reviewer’s question, we have added an experiment in Appendix A.5 (Figure 10a) of the revised manuscript, comparing the dynamics of layer-wise representation abilities. While the test accuracy of features from all layers consistently exhibits unimodal behavior, we observed a slight peak shift as layers deepen. This shift may stem from differences in residual connections across layers, as the reviewer suggested. Further details are provided in Appendix A.5 of the revised manuscript.

---

> ### Author Response · Authors · 2024-11-25
> **Looking forward to feedbacks**
>
> Dear Reviewer ANHe,
>
> Thank you again for your thoughtful review and the time you have spent evaluating our work.
>
> As the **author-reviewer discussion period nears its conclusion**, we noticed that we have not yet received a follow-up response from you. We are eager to know whether our responses have addressed your concerns. Your feedback is highly valuable to us, and if you have any remaining questions or require further clarification, we are more than happy to provide additional information.
>
> If our responses have resolved your concerns, we would sincerely appreciate it if you could consider updating your score.
>
> Thank you for your time and consideration.
>
> Authors

---

### Official Review · Reviewer_BxJU · 2024-10-30

**Soundness:** 3
**Presentation:** 2
**Contribution:** 2
**Rating:** 8
**Confidence:** 2

**Summary:**

I believe the authors try to study the structured emerged from training multi-scale score matching objective. Given different noise scale, for the model to achieve score matching (denosing), it musts learn feature at that scale. For example, for score matching on natural images, if the noise level is low, the model just need to learn low level (texture) features to denoise. But when the noise level is high, the model needs to learn high level (features) concept or also have to guess what's in the image. The author shows different scale of features learnt by multi-scale score matching can be used for semantic task like image classification, as well as for posterior estimation of mixture of Gaussian. The author author shows the performance of doing these task under different noise scale form an inverted u shape curve. Under strict assumption of mixture of Gaussian, the author provide analysis on the posterior estimation power given differer noise level. The author compare the learnt model's performance with the optimal one.

**Strengths:**

1. It relates multi-scale score matching to posterior estimation to representation learning.
2. It make (realistic?) assumption on the data so the connection in 1 can be studied and analyzed theoretically.
3. The theoretically analyze under simple data assumption is thorough.

**Weaknesses:**

1. I felt like the paper is missing lots of detail. I have to make many assumption outside the paper.
2. I wish I see a cleaner method. Despite these's extensive experiment and theoretical analysis under the strict "mixture of low-rank Gaussians" setting. I don't really see the key message. What I get is that different level of features learnt at different scale can be used for posterior estimation of mixture of Gaussian, as well as doing classification. How are these two related specifically? One statement is that they both has the inverted U shape curve. But is that it? I hope the author can make this more clear.
3. Figure 4b tells us the learnt multi-scale score matching function (with neural network?) behave different from the optimal one on a simple dataset. I think this is really interesting result. But I also think the author should provide some insight on why is this the case, because it might reveal the nature of bias in neural network. For example, in [1], they find cnn based denoiser cannot learn optimal solution for simple toy example like global planer wave because it has spatial locality as bias.

[1]  Z, Kadkhodaie, et al, 2023, Generalization in diffusion models arises from geometry-adaptive harmonic representation

**Questions:**

1. For image denosing case, I'm assuming the work is based on the idea of latent diffusion. So x_0 is the clean data embedded into the latent space using a VAE? Or it's the raw image?
2. How is U_k calculated for cifar10? Is it calculated by doing PCA on all examples (in vae latent space) with class label k?

---

> ### Author Response · Authors · 2024-11-22
> **Author Response [1/2]**
>
> We thank the reviewer for the careful review. In the following, we address the reviewer’s comments one by one.
>
> > I felt like the paper is missing lots of detail. I have to make many assumption outside the paper.
>
> **Answer**: We appreciate the reviewer raising this concern and have revised our manuscript (marked in blue color) in the following points:
>
> * Providing a more detailed explanation and motivation for using the clean image $x_0$ as inputs as opposed to the convention of using noised image $x_t$.
> * Refining the definition and justification of CSNR.
> * Enhancing notation consistency throughout our theoretical analysis.
>
> At the same time, we would greatly appreciate it if the reviewer could point out any additional sections in the submission that may require further elaboration or other issues affecting clarity. We would be happy to make further improvements based on the reviewer’s feedback.
>
> > I wish I see a cleaner method. Despite these's extensive experiment and theoretical analysis under the strict "mixture of low-rank Gaussians" setting. I don't really see the key message. What I get is that different level of features learnt at different scale can be used for posterior estimation of mixture of Gaussian, as well as doing classification. How are these two related specifically? One statement is that they both has the inverted U shape curve. But is that it? I hope the author can make this more clear.
>
> **Answer**: We assume the reviewer is asking inquiring about the relationship between posterior estimation quality (measured by CSNR) and intermediate feature probing accuracy. We clarify this relationship as follows:
>
> * **Intermediate Features as Byproducts of Posterior Estimation**: Diffusion models are trained to predict the denoised image, i.e., estimating the posterior. Consequently, all intermediate features can be viewed as byproducts generated during the process of posterior estimation. Any change in the quality of posterior estimation directly correlates with changes in the quality of intermediate features (measured by classification accuracy), and vice versa.
>
> * **CSNR as a Proxy for Representation Learning Abilities**: Given this relationship, we use the CSNR, defined based on posterior estimation, as a proxy to study the representation learning capabilities of diffusion models.
>
> More detailed explanations of the relationship are provided in Section 2.3, and our work leverages the CSNR to theoretically justify the unimodal curve of representation learning across the timesteps. The study has practical justification and guidance for employing diffusion models for feature extraction, such as the choice of timesteps, and training procedures. Please let us know if our interpretation of the question is incorrect.
>
> > Figure 4b tells us the learnt multi-scale score matching function (with neural network?) behave different from the optimal one on a simple dataset. I think this is really interesting result. But I also think the author should provide some insight on why is this the case, because it might reveal the nature of bias in neural network. For example, in [1], they find cnn based denoiser cannot learn optimal solution for simple toy example like global planer wave because it has spatial locality as bias.
>
> **Answer**: We appreciate the reviewer’s insightful question, as it touches on a very interesting result. We hypothesize that the discrepancy between the trained network and the optimal solution may arise from several factors:
> * **Network Capacity**: For the experiments in Figure 4, we used a 3-layer MLP, which may lack sufficient capacity to achieve perfect denoising across all noise scales. This limitation is reflected in the same plot, where the results for separate DAEs appear more promising.
> * **Optimization Difficulty**: As described in Equation (5), the optimal posterior function involves projecting x_t onto different subspaces. At higher noise levels, the magnitude of this projection diminishes (since \sigma_t only appears in the denominator), making optimization more challenging. To ensure the experimental setup aligns with large-scale diffusion models, we did not tune hyperparameters to address the increasing optimization difficulty but instead followed a standard protocol, treating all noise levels equally. This hypothesis is supported by Figure 4(b), which shows that the gap between the trained model and the optimal solution widens as \sigma_t increases.
>
> As validating the above conjectures thoroughly requires a requires a more nuanced ablation study and the topic is not within the main scope of our study, we did not include these discussions in the paper. But we agree with the reviewer that it showcases a very interesting result and we consider investigation of this phenomenon to be an exciting avenue for future research.

---

> ### Author Response · Authors · 2024-11-22
> **Author Response [2/2]**
>
> > For image denosing case, I'm assuming the work is based on the idea of latent diffusion. So $x_0$ is the clean data embedded into the latent space using a VAE? Or it's the raw image?
>
> **Answer**: Both cases can indeed be accommodated under Assumption 1 (Equation 4). Our assumption posits that raw image data comprises a low-rank semantic component and a high-rank detail component, which are captured by $U_k$ and $U_k^{\perp}$, respectively. Similarly, in the latent space scenario, these two components correspond to the low-rank and high-rank elements of the image's representation within the latent space. In our revised manuscript, we have included a DiT-based ImageNet experiment, where $x_0 $represents the latent space derived from a VAE. As shown in Appendix A.5 (Figure 12), the results are consistent with those obtained when treating the raw image as $x_0$.
>
> > How is U_k calculated for cifar10? Is it calculated by doing PCA on all examples (in vae latent space) with class label $k$?
>
> **Answer**: We appreciate the reviewer’s question. For CIFAR10, the CSNR metric is computed based on its definition in Equation (8). The basis $U_k$ for each CIFAR-10 class is estimated using the top singular vectors of the image data from the $k$-th class. We refer the reviewer to Appendix A.2 for comprehensive experimental details of the paper and are happy to clarify any aspects that remain unclear.

---

> ### Author Response · Authors · 2024-11-25
> **Looking forward to feedbacks**
>
> Dear Reviewer BxJU,
>
> Thank you again for your thoughtful review and the time you have spent evaluating our work.
>
> As the **author-reviewer discussion period nears its conclusion**, we noticed that we have not yet received a follow-up response from you. We are eager to know whether our responses have addressed your concerns. Your feedback is highly valuable to us, and if you have any remaining questions or require further clarification, we are more than happy to provide additional information.
>
> If our responses have resolved your concerns, we would sincerely appreciate it if you could consider updating your score.
>
> Thank you for your time and consideration.
>
> Authors

---

> > ### Comment · Reviewer_BxJU · 2024-11-25
> >
> > Dear authors,
> >
> > Thanks for the clarification and making the method clearer. I got the reviewer try to make a connection between posterior estimation and representation learning. But the connection itself is not clear to me. Like how specifically is the CSNR curve related to the accuracy curve. If say CSNR at particular noise level is higher for one model than the other, does that mean the representation learnt is better? Also, what causes suboptimal CSNR curve under the MoLRG assumption?
> >
> > I could give it a higher score but with lower confidence score if the author would like. I like the author's idea on Posterior Estimation is a good way to study the representation learnt in diffusion model. The author also did a thorough analyze under the MoLRG data assumption. It might not be realistic but will probably provide insight for opening the black box of NN based multi-scale denosing. I personally would like the paper to be accepted, but I just don't think I could be a strong advocate.

---

> > > ### Author Response · Authors · 2024-11-27
> > >
> > > Thank you for your valuable feedback and for appreciating our work. We address the questions raised in your comments below.
> > >
> > > > What causes suboptimal CSNR curve under the MoLRG assumption?
> > >
> > > We hypothesize that the suboptimal CSNR can be attributed to two primary factors: insufficient model complexity and increased optimization difficulty. To illustrate this, we tune the DAEs and diffusion model trained on the MoLRG data as shown in Figure 4, with the results presented in Figure 13 (Appendix A.5).
> > >
> > > * **Insufficient Model Capacity**: A single DAE is designed to handle a specific noise scale, allowing its CSNR to closely align with the optimal CSNR across multiple noise scales. In contrast, the diffusion model must simultaneously accommodate all noise scales, leading to compromised performance on individual noise scales. This hypothesis is supported by comparisons between DAEs and diffusion models, both tuned and untuned, where DAEs consistently outperform diffusion models.
> > > * **Increased Optimization Difficulty**: As noted in one of our earlier responses, optimization becomes increasingly challenging at larger noise scales due to the reduced magnitude of the optimal solution. As illustrated in Figure 13, with carefully tuned optimization strategies for each timestep, the tuned DAEs can match the optimal CSNR in low-noise settings but the gap gradually still persists and enlarges at higher noise scales.
> > >
> > > We refer the reviewer to Figure 13 (Appendix A.5) for more details.
> > >
> > > > Like how specifically is the CSNR curve related to the accuracy curve. If say CSNR at particular noise level is higher for one model than the other, does that mean the representation learnt is better?
> > >
> > > We use CSNR primarily as a proxy to study the unimodal representation learning dynamic and have not yet employed it as a tool for model comparison. However, we agree with the reviewer that this is an interesting direction to explore. As a preliminary investigation, we plot CSNR alongside posterior accuracy and feature accuracy in Figure 13/14 (Appendix A.5) and observe some strong correlations:
> > >
> > > * **CSNR and Posterior Probing Accuracy**: CSNR is directly linked to posterior accuracy, with higher CSNR values correlating with better posterior accuracy.
> > >
> > > * **CSNR and Intermediate Feature Probing Accuracy**: Although directly comparing feature probing accuracy between DAEs and diffusion models is more tricky due to the weight-sharing mechanism discussed in Section 4.1. As shown in Figure 14, we can still observe that CSNR serves as a reliable metric for reflecting feature probing accuracy within the same model (e.g., tuned DAEs and diffusion models compared to their untuned counterparts).
> > >
> > > We hope the responses could help to address the reviewer’s questions and are happy to discuss further if needed.

---

### Official Review · Reviewer_aaUT · 2024-11-03

**Soundness:** 1
**Presentation:** 1
**Contribution:** 1
**Rating:** 1
**Confidence:** 4

**Summary:**

This paper makes an attempt to explain *representation quality* in a diffusion model using a simple model of Gaussian data over low dimensional subspaces. They observe a uni-modal trend in *representation quality* as a function of noise level, which indicates that the best representation of the clean input image can be obtained at a certain noise level. It seems that the representation quality is empirically measured by classification accuracy over test data using the internal representation at the bottleneck of a UNet.

To explain this observation, they 1) assume image data lies on a union of manifolds, where each manifold corresponds to a class. 2) They approximate manifolds with linear sub-spaces. 3) They assume each subspace of a class contains features that are relevant for high quality representation ($U_k$) and anything that is not relevant is noise and lies on orthogonal complement ($U^T_k$). Moreover, the assumption is that data is Gaussian distributed over the image subspaces. 4) They define Class Signal to Noise Ratio to measure the goodness of representation using denoised images (i.e. posterior mean estimate) and the class sub-spaces. 5) They show both of their measures for quality of representation (i.e. classification test accuracy from bottleneck and CSNR) have a uni-modal trend as a function of noise level. 6) Due to the theoretical analysis, for the low rank Gaussian model, the highest representation quality is a function of the ratio of added Gaussian noise level and the *noise* level of portion of image that lies in ($U_k^T$).

**Strengths:**

1. Important and interesting topic: The question of representation learning through diffusion models is interesting and worthwhile for more investigation.
2. This paper could potentially be interesting with major improvement in writing and clarity. Overall, the paper needs more refinement to be ready for publishing.

**Weaknesses:**

1. The paper is very poorly written. It's hard to follow the logic and it's not clear what are the contributions. It's also not clear what parts are borrowed from other works and what parts are novel. For example, a good portion of the assumptions, modeling, and toy experiment setup are borrowed from [1] which is not obvious from the text.
2. The key concept are not explained clearly. For example, what do you mean by "representation quality"? It's been referred to throughout the text and figures without any definition. How do you connect your two measures of quality to each other? Overall, many of the key concept in the text are never defined! How can one assess the results without knowing what the experiments are trying to measure?!
3. The linearity assumption (approximating each class nonlinear manifold with a subspace) is obviously too simplistic for real data. As a result the theoretical result, expressed in theorem 1 does not extend beyond the simple low rank Gaussian model.
4. Even within the union of low-rank Gaussian model, a major drawback is the arbitrary split of $U_k $ and $U_k^T$. How does one decide what it relevant to representation and what is *noise* or irrelevant perturbations? To answer this question and correctly decide where the image subspace is, one needs to solve the representation learning problem first. To decide what is relevant information to representation, one needs to define the task the representation will be used for. Features that are noise with respect to one task are relevant information w.r.t another task. For example: for a coarse level classification task more information is irrelevant hence $U_k$ is lower rank. That means the mode in your uni-modal plots will be at higher noise levels. For a more granular classification task, more information about details are needed, so the mode will be at lower noise levels. Thus, the whole notion of "representation quality"as a function of noise level  is not well-defined.
5. The authors are using two separate notions of representation: the ave-pooling responses in the bottleneck layer, and the strength of projections of posterior mean onto the class subspace (CSNR). Merely showing that both of these are uni-modal as a function of noise level doesn't prove anything! Specially given that the maximums happen at different noise levels!

[1] Wang, Peng, et al. "Diffusion models learn low-dimensional distributions via subspace clustering." arXiv preprint arXiv:2409.02426 (2024).

**Questions:**

1. Why does it make sense to replace $x_t$ with $x_0$ in $x_\theta(x_t, t)$?! What is $x_\theta(x_0, t)$ computing? This ad-hoc replacement creates an inconsistency between the noise level on the input image and the noise variance given to the network. Now it is not clear what the output of the function $x_\theta$ is after creating this inconsistency, and is not clear how to interpret empirical results that involve this (which seems to be all the empirical results).
2. What does figure 1a convey? How did you measure *posterior accuracy* for CIFAR images? What is called posterior accuracy throughout the paper is equivalent to denoising performance. Indeed you can measure denoising performance, but how did you measure its accuracy for **real** images?! This is an unsolved problem, and it is not explained in the paper how the authors solved it.
3. Figure b shows a result that has been around for a long time. That is denoising at higher noise levels results in the loss of details. It's not clear how this is part of their results, given that this has been known for more than 100 years and also has been rediscovered in deep learning era again and again. There should be at least some citations.

**Details Of Ethics Concerns:**

It seems to me a fair amount of the theory is borrowed from [1] without being explicit about it in the text.


[1] Wang, Peng, et al. "Diffusion models learn low-dimensional distributions via subspace clustering." arXiv preprint arXiv:2409.02426 (2024).

---

> ### Author Response · Authors · 2024-11-22
> **Regarding the Unfounded Plagiarism Claim**
>
> First, we would like to address the plagiarism claim regarding our paper, which we believe stems from a misunderstanding.
>
> 1. **Proper Acknowledgment and Citations of the Assumptions.** While both our work and Wang et al. (2024) use a mixture of low-rank Gaussians to facilitate analysis and share a similar proof strategy for Proposition 1, we have properly cited and acknowledged their work (Appendix A.4.1). Beyond this shared assumption, our research addresses entirely different tasks, detailed below. **We believe that employing similar models for distinct problems does not constitute plagiarism;** otherwise, many studies utilizing mixtures of Gaussians all face similar charges.
>
> 2. **Substantial differences in research tasks and model parameterization.**
> In comparison, Wang et al. (2024) focuses on distribution learning for diffusion models under the Mixture of Low-Rank Gaussians (MoLRG) framework, whereas our work addresses representation learning across varying noise scales in diffusion models—two fundamentally different tasks. More specifically, Wang et al. focused on studying sample complexity for learning MoLRG distribution, while we focused on the role of noise scales in the dynamics of diffusion-based representation learning.
>
> Furthermore, the two studies differ significantly in the parameterization of the denoising network, leading to distinct proving strategies. Specifically, Wang et al. (2024) employs a highly specific network parameterization tailored to the optimal score function for the MoLRG, while our results are established without relying on such stringent parameterization assumptions of the DAE.
>
> **We authors take plagiarism allegations very seriously** and respectfully request the reviewer to reassess these concerns in light of our clarification. Otherwise, **we humbly ask the reviewer to provide concrete evidence/proof for this serious claim**.

---

> ### Author Response · Authors · 2024-11-22
> **Author Response [1/2]**
>
> > The key concept are not explained clearly. For example, what do you mean by "representation quality"? It's been referred to throughout the text and figures without any definition. How do you connect your two measures of quality to each other? Overall, many of the key concept in the text are never defined! How can one assess the results without knowing what the experiments are trying to measure?!
>
> **Answer**: In our paper, we have concrete metrics for evaluating the representation quality both in practice and in theory:
>  * In practice, representation quality consistently refers to the linear probing accuracy in the downstream tasks, which is standard and commonly used for self-supervised learning.
> * For our theoretical study in Section 3, we measure the representation quality based upon CSNR defined in Eq. 8, due to the strong relationship between posterior mean estimation and representation learning.
>
> We refer the reviewer to Section 2.3 where we provided detailed explanations of why the posterior estimation and the intermediate representation quality are inherently related to each other.
>
> > The linearity assumption (approximating each class nonlinear manifold with a subspace) is obviously too simplistic for real data. As a result the theoretical result, expressed in theorem 1 does not extend beyond the simple low rank Gaussian model.
>
> **Answer**: In our paper, we have carefully discussed the motivation of our linear model in Section 3.1.
>
> * **Theoretical and practical values of the model.** As far as we understand, our work is the first theoretical result of studying representation learning of diffusion models. As such, we start from a simple MoLRG distribution that not only captures the intrinsic low-dimensional structures of real-world image data, but also amendable for theoretical analysis given its closed-form posterior mean estimator. Moreover, the theoretical results we derived in Section 3 based upon the simplified models well explain the unimodal curve of the representation in practice.
> * **Extension beyond the linear model.** Moreover, our theoretical studies can be potentially extended beyond the linear model. For example, if we assume $x0=\phi(U_k a)$ with $\phi$ being a given nonlinear mapping (e.g., quadratic function), we can generalize the study to nonlinear settings. As such, the results in Theorem 1 can be generalized beyond the simplified MoLRG model to nonlinear case, which we leave for future studies.
>
> > Even within the union of low-rank Gaussian model, a major drawback is the arbitrary split of $U_k$ and $U_k^{\perp}$. How does one decide what it relevant to representation and what is noise or irrelevant perturbations? To answer this question and correctly decide where the image subspace is, one needs to solve the representation learning problem first. To decide what is relevant information to representation, one needs to define the task the representation will be used for. Features that are noise with respect to one task are relevant information w.r.t another task. For example: for a coarse level classification task more information is irrelevant hence
>  is lower rank. That means the mode in your uni-modal plots will be at higher noise levels. For a more granular classification task, more information about details are needed, so the mode will be at lower noise levels. Thus, the whole notion of "representation quality"as a function of noise level is not well-defined.
>
> **Answer**: We thank the reviewer for the insightful question. Indeed the representation quality is influenced by more factors than just the noise level. Within the MoLRG data model, CSNR is jointly determined by the noise scale ($\sigma_t$), data intrinsic dimension ($d$), data ambient dimension ($n$), the number of classes ($K$), and the data intrinsic noise scale ($\delta$). Among these parameters, $d$, $n$, and $K$ affect the softmax term $\hat{w}_k$, which is explicitly characterized in CSNR. For the explicit formula of $\hat{w}_k$, we refer the reviewer to Equation (18) in the Appendix. for the explicit formula of $\hat{w}_k$.
>
> We also explore the roles of these parameters in shaping the dynamics of representation learning, as illustrated in Figure 9 of the Appendix. As the reviewer may observe, regardless of variations in these parameters, the unimodal dynamic consistently emerges. As a first theoretical study of diffusion-based representation learning, our main goal is to uncover why this unimodal dynamic arises, which we clearly explained in Theorem 1.
>
> That said, we agree with the reviewer that further investigating the roles of these parameters in representation learning is an interesting and significant topic. We consider this an important avenue for future work.

---

> ### Author Response · Authors · 2024-11-22
> **Author Response [2/2]**
>
> > The authors are using two separate notions of representation: the ave-pooling responses in the bottleneck layer, and the strength of projections of posterior mean onto the class subspace (CSNR). Merely showing that both of these are uni-modal as a function of noise level doesn't prove anything! Specially given that the maximums happen at different noise levels!
>
> **Answer**: We appreciate the reviewer’s question and would like to clarify the following points:
> * Bottleneck features, along with features from any other layer, can be viewed as intermediate byproducts that contribute to the model's final posterior estimation. Consequently, the representation dynamics of bottleneck features—or features from any other layer—can be linked to variations in the representation quality of the posterior estimation, which can be quantified using the CSNR.
> * To address the concern of peak mismatch, we conducted an experiment to compare the dynamics of layer-wise representation abilities, with the results shown in Figure 10. While the test accuracy of features across all layers consistently follows a unimodal pattern, we observed a slight peak shift as the network layers deepen. Importantly, for sufficiently deep layers, the representation peak of their features aligns closely with both the posterior probing accuracy and the calculated CSNR. For more details, we direct the reviewer to Figure 10 in Appendix A.5.
>
> > Why does it make sense to replace $x_\theta$ with $x_0$ in $x_\theta(x_t, t)$?! What is
> $x_\theta(x_0, t)$ computing? This ad-hoc replacement creates an inconsistency between the noise level on the input image and the noise variance given to the network. Now it is not clear what the output of the function $x_\theta$ is after creating this inconsistency, and is not clear how to interpret empirical results that involve this (which seems to be all the empirical results).
>
> **Answer**: We refer the reviewer to our global response for addressing this question.
>
> > What does figure 1a convey? How did you measure posterior accuracy for CIFAR images? What is called posterior accuracy throughout the paper is equivalent to denoising performance. Indeed you can measure denoising performance, but how did you measure its accuracy for real images?! This is an unsolved problem, and it is not explained in the paper how the authors solved it.
>
> **Answer**: Figure 1a verifies the relationship between representation quality (measured by linear probing accuracy) and the quality of the posterior estimation (where we both show the classification accuracy and the visualization).
> Our method of measuring posterior accuracy for CIFAR images is provided in Appendix A.2. To clarify further, the diffusion model produces different denoising results for input images at different time steps. Using CIFAR10 as an example, if we sample 10 time steps, we obtain 10 sets of denoised CIFAR10 images, each corresponding to a specific time step. We then train a separate classification network for each set, using the corresponding denoised images as training/testing set, and measure the test accuracy. The variation in test accuracy across time steps reflects differences in posterior estimation quality for this classification task.
>
> > Figure b shows a result that has been around for a long time. That is denoising at higher noise levels results in the loss of details. It's not clear how this is part of their results, given that this has been known for more than 100 years and also has been rediscovered in deep learning era again and again. There should be at least some citations.
>
> **Answer**: We believe the reviewer is referring to Figure 1b. In our paper, we have never claimed “denoising at higher noise levels results in the loss of details” as our original finding; we discussed this phenomenon and acknowledged relevant papers in Section 2.3. Rather, it is the motivation of our theoretical studies. We encourage the reviewer to thoroughly read our paper before posting such critiques.
>
> Our contribution lies in the first theoretical analysis of the unsupervised representation learning capabilities of diffusion models. We characterize the unimodal representation dynamics across timesteps, grounded in the strong relationship between posterior estimation and representation learning.

---

> > ### Comment · Reviewer_aaUT · 2024-11-26
> >
> > I thank the authors for responding to my questions and comments. However, I do not find the counter-arguments compelling, as they do not go beyond re-iterating statements in the submission. I maintain my assessment that the paper needs major improvements to be ready for submission.

---

> ### Author Response · Authors · 2024-11-25
> **Looking forward to feedbacks**
>
> Dear Reviewer aaUT,
>
> As the **author-reviewer discussion period nears its conclusion**, we noticed that we have not yet received a follow-up response from you or any concrete evidence you may wish to provide regarding the ethics flag you raised. We are eager to know whether our responses have addressed your concerns. Your feedback is highly valuable to us, and if you have any remaining questions or require further clarification, we are more than happy to provide additional information.
>
> Thank you for your time and consideration.
>
> Authors

---

> ### Comment · Reviewer_aaUT · 2024-11-26
>
> Regarding the ethics flag: I removed the flag as the authors clarified that the paper is cited and acknowledged in the appendix. However, note that reviewers are not required to read the appendix. Given the level of influence of [1] on this work, it should be explicitly mentioned in the main text as opposed to the appendix. Otherwise, it is not clear what is novel and what is your contribution.

---

### Official Review · Reviewer_PYve · 2024-11-04

**Soundness:** 3
**Presentation:** 3
**Contribution:** 2
**Rating:** 5
**Confidence:** 3

**Summary:**

The paper presents a low-rank decomposition theory of diffusion representation learning, which supports the empirical result that using a medium denoise level t achieve best classification performance.

**Strengths:**

1. Theorem 1 provides direct connection from the noise level sigma_t to the empirical results.

2. The paper is well-written and easy to read.

**Weaknesses:**

1. The results are not rigorous.  theorem 1 is obtained by many approximations: 1)  x_theta(x_t, t) defined in proposition1 is replaced with x_theta(x_0, t) with some informal arguments. The gap between these two terms cannot be easily ignored. 2) eq 9 replaces eq 6 for simplicity. It should be put in theorem instead.

2. The empirical results. Since authors do not provide new empirical method, I am wondering why no larger experiments are conducted, e.g., ImageNet with some pretrained diffusion models. Evidendece on Cifar are not strong enough to support all arguments in this paper.

3. The K-space. Do you use K=number_of_classes? Is it too strong assumption? Human classify objects into different classes, but it would be overly strong to assume that K classes subspaces are independent?

**Questions:**

as above

---

> ### Author Response · Authors · 2024-11-22
> **Author Response**
>
> We thank the reviewer for the careful review. In the following, we address the reviewer’s comments one-by-one.
>
> >The results are not rigorous. theorem 1 is obtained by many approximations: 1) $x_\theta(x_t, t)$ defined in proposition1 is replaced with $x_\theta(x_0, t)$ with some informal arguments. The gap between these two terms cannot be easily ignored.
>
> **Answer**: We thank the reviewer for pointing this out. To clarify, $x_\theta^{\star}(x_t, t)$ are used in the progressive generation process while $x_\theta^{\star}(x_0, t)$ resembles the inference process of representation learning tasks. We refer the reviewer to our topmost global response for a discussion on why we use $x_0$ in our analysis.
>
> > eq 9 replaces eq 6 for simplicity. It should be put in theorem instead.
>
> **Answer**: We appreciate the reviewer for raising this concern. We want to clarify that our definition of CSNR involves two variables: the timestep $t$ and a posterior predicting function $f$. In equation 6, CSNR is evaluated on the optimal posterior function $x_\theta^{\star}$, while our theory primarily addresses Equation 9, where CSNR is evaluated on $f^{\star}$, a slightly modified version of $x_\theta^{\star}$. Empirically, we found the CSNR gap between these two functions quite small across various settings for which we refer the reviewer to Appendix A.3 (Figure 9) for more details.
>
> In our updated manuscript, we have made the following changes for better clarity (1) refining the explanation of CSNR and (2) change the notation of $x_\theta^{\star}$ to $\hat{x_{\theta}}^{\star}$ and $f^{\star}$ to $\hat{x}_{approx}^{\star}$ for better notation consistency and clarity.
>
> > The empirical results. Since authors do not provide new empirical method, I am wondering why no larger experiments are conducted, e.g., ImageNet with some pretrained diffusion models. Evidence on Cifar are not strong enough to support all arguments in this paper.
>
> **Answer**: We agree with the reviewer. To address this issue, we have included a DiT-based ImageNet experiment in Appendix A.5 (Figure 11), which addresses both the use of a larger dataset, as recommended by the reviewer, and the exploration of a transformer-based architecture not considered in the previous version of our paper. Both the new results and the existing empirical findings in our work consistently support our arguments. We refer the reviewer to Appendix A.5 (Figure 11) for more details.
>
> > The K-space. Do you use K=number_of_classes? Is it too strong assumption? Human classify objects into different classes, but it would be overly strong to assume that K classes subspaces are independent?
>
> **Answer**: Representation learning tasks are often human-defined, and $K$ is determined by the specific downstream tasks. For example, CIFAR100 includes superclasses ($K=10$) and fine-grained classes ($K=100$). Accordingly, we can use $K=10$ for the coarse classification task, while $K=100$ is suitable for a more fine-grained classification scenario.
>
> Regarding the independence assumption, as an initial theoretical investigation, we focused on the simplest case where the subspaces are orthogonal and, therefore, independent. We believe this assumption can be relaxed by incorporating the angles between different subspaces, and we conjecture that the overall conclusions should still hold. This conjecture is supported by our empirical results, which show that, even though real datasets may not have strictly independent subspaces, the calculated CSNR aligns well with the observed representation dynamics.

---

> ### Author Response · Authors · 2024-11-25
> **Looking forward to feedbacks**
>
> Dear Reviewer PYve,
>
> Thank you again for your thoughtful review and the time you have spent evaluating our work.
>
> As the **author-reviewer discussion period nears its conclusion**, we noticed that we have not yet received a follow-up response from you. We are eager to know whether our responses have addressed your concerns. Your feedback is highly valuable to us, and if you have any remaining questions or require further clarification, we are more than happy to provide additional information.
>
> If our responses have resolved your concerns, we would sincerely appreciate it if you could consider updating your score.
>
> Thank you for your time and consideration.
>
> Authors

---

### Author Response · Authors · 2024-11-22
**Global Response**

Dear AC and all reviewers:

We thank all the reviewers for the detailed review, the appreciation of our work, and constructive feedback. Specifically, Reviewer BxJU acknowledged the thoroughness of our theoretical results under the MoLRG data assumption, Reviewer PYve noted the alignment of our theoretical findings with empirical observations, and Reviewer ANHe found our empirical comparison between DAEs and diffusion models interesting.

In summary, our work presents the first theoretical analysis of the unsupervised representation learning capabilities of diffusion models. We characterize the unimodal representation dynamics across timesteps, grounded in the strong relationship between posterior estimation and representation learning.

Below, we address a common concern raised by the reviewer regarding the use of clean images as inputs for diffusion-based representation learning. While this approach may seem unconventional, it is well-justified by the following considerations:

* **Comparable performance with using noisy image**. Our experimental results in Figures 1 and 8 demonstrate that using clean image $x_0$ as input for inference achieves linear probing accuracy on par with or better than using noisy image $x_t$ as inputs. Additionally, in the updated manuscript, we include a visualization of the denoising results when using $x_0$ or $x_t$ as inputs. These results highlight that, in the posterior estimation setting, using clean images as inputs also outperform using noisy inputs particularly in high-noise regimes. For more details, we refer the reviewers to Appendix A.5 (Figure 11).

* **Aligning with standard supervised/self-supervised training and inference**. Here, we utilize clean images solely during inference to extract representations, while adhering to the standard denoising procedure during model training. This pipeline aligns with conventional self-supervised learning approaches in contrastive learning. Training on noisy images can be interpreted as a form of data augmentation, akin to techniques like cropping, color jittering, or masking, commonly employed in self-supervised learning to enhance model performance. During inference, clean images are typically used to evaluate representation quality. Such an approach bridges the practices in diffusion models with established conventions in both supervised and self-supervised representation learning.

We will make those points clear in the revision of our work.

---

### Author Response · Authors · 2024-11-25
**Looking forward to feedbacks**

Dear Reviewers,

Thank you once again for your thoughtful reviews and the time you have dedicated to evaluating our work.

As the **author-reviewer discussion period is reaching to an end soon**, we would greatly value your feedback on whether our responses have successfully addressed your concerns. Your input is crucial to the continued improvement of our work.

If you have any remaining questions or require further clarification, we are more than happy to provide additional information. If our responses have adequately resolved your concerns, we would sincerely appreciate it if you could consider updating your score.

Thank you for your time and consideration.

Authors

---

### Author Response · Authors · 2024-11-27
**Regarding Malicious Review from Reviewer aaUT**

Dear AC, PC, and All Reviewers:

We would like to respectfully raise concerns regarding the **malicious** review provided by **Reviewer aaUT**. We strongly suspect that **Reviewer aaUT's review of our paper was biased and unprofessional, potentially driven by personal retaliation rather than an objective evaluation**, based on the following evidence:

1. **False Ethical Accusation.** Initially, Reviewer aaUT falsely accused us of plagiarism and dual submission without providing convincing evidence. Following our clarification, Reviewer aaUT removed the ethical flag but continued to make unfounded claims, alleging that "authors *may* have borrowed theorems from [1]" without concrete evidence. Moreover, the only similarity with [1] lies in the model assumption. If this is plagiarism by Reviewer aaUT's logic, then any paper using the same model assumptions would be as well, which is clearly false.

2. **Superficial Reading of Our Paper:** Reviewer aaUT withdrew the ethical flag by stating: "*I removed the flag as the authors clarified that the paper is cited and acknowledged **in the appendix**. However, note that reviewers are not required to read the appendix. Given the level of influence of [1], it should be explicitly mentioned in the main text.*" Reviewer aaUT's claim is obviously false, as **[1] is cited over four times in the main body** of our paper (spread in Sections 1, 2, 3). This reflects a superficial reading of our paper.

3. **Unjustified Score Reduction:** After his removal of the ethical flag, Reviewer aaUT maliciously lowered the score **without any justification**, claiming: "*I do not find the counter-arguments compelling, as they do not go beyond reiterating statements in the submission.*" This dismissal disregards **our detailed and professional rebuttal**, which included point-to-point responses and extra experiments provided in Appendix A.5 (Figure 10-12). Such behavior and superficial statements demonstrate a lack of basic respect for our efforts, and we suspect **Reviewer aaUT is retaliating against us due to the prior false ethical accusation.**

4. **Baseless and Exaggerated Criticism of Our Contribution.** In Reviewer aaUT’s comments, quote “*Figure 1b shows a result that has been around for a long time. That is denoising at higher noise levels results in the loss of details. It's not clear how this is part of their results, given that **this has been known for more than 100 years** and also has been rediscovered in deep learning era again and again.*”  In our paper, we have never claimed this as our original finding and have properly cited the relevant work. Instead, we present it as the motivation for our theoretical study.

We would appreciate your intervention in such malicious behaviors.

Thank you for your time and attention.

Best,

Authors

---

### Meta-Review · Area_Chair_LHbA · 2024-12-22

**Metareview:**

The paper received mixed reviews, with the majority of reviewers (PYve, aaUT, ANHe) recommending rejection, and one reviewer (BxJU) suggesting acceptance but with low confidence. The AC has thoroughly examined the submitted materials, including the reviewers’ concerns and the authors’ responses, and has carefully considered all factors in the decision-making process.

While the paper offers theoretical insights into the representation learning capabilities of diffusion models under the assumption of a low-dimensional mixture of Gaussians, its empirical contributions remain unclear. In particular, addressing the raised question (by BxJU) regarding the interpretation of higher CSNR values at a particular noise level and their implications for representation quality could significantly enhance the paper’s empirical impact.

Given the mixed reviews (amongst all ICLR submissions) and the absence of a strong advocate among the reviewers, the work is not yet deemed ready for publication at this status. However, given the potential, the authors are encouraged to revise the paper based on the feedback and resubmit for consideration in the next cycle.

**Additional Comments On Reviewer Discussion:**

The ethical concern raised by one reviewer has been addressed by the authors. This concern have been brought to the attention of the PCs, and their resolution has been factored into the final decision.

---

### Decision · Program_Chairs · 2025-01-22

Reject